# L²ight: Enabling On-Chip Learning for Optical Neural Networks via Efficient *in-situ* Subspace Optimization

**Jiaqi Gu, Hanqing Zhu, Chenghao Feng, Zixuan Jiang, Ray T. Chen, David Z. Pan**
ECE Department, University of Texas at Austin
*{jqgu, hqzhu,fengchenghao1996,zixuan}@utexas.edu, {chen, dpan}@ece.utexas.edu*

## Abstract

Silicon-photonics-based optical neural network (ONN) is a promising hardware platform that could represent a paradigm shift in efficient AI with its CMOS-compatibility, flexibility, ultra-low execution latency, and high energy efficiency. *In-situ* training on the online programmable photonic chips is appealing but still encounters challenging issues in on-chip implementability, scalability, and efficiency. In this work, we propose a closed-loop ONN on-chip learning framework L²ight to enable scalable ONN mapping and efficient *in-situ* learning. L²ight adopts a three-stage learning flow that first calibrates the complicated photonic circuit states under challenging physical constraints, then performs photonic core mapping via combined analytical solving and zeroth-order optimization. A subspace learning procedure with multi-level sparsity is integrated into L²ight to enable *in-situ* gradient evaluation and fast adaptation, unleashing the power of optics for real on-chip intelligence. Extensive experiments demonstrate our proposed L²ight outperforms prior ONN training protocols with **3-order-of-magnitude** higher scalability and over **30×** better efficiency, when benchmarked on various models and learning tasks. This synergistic framework is the *first* scalable on-chip learning solution that pushes this emerging field from *intractable* to *scalable* and further to *efficient* for next-generation self-learnable photonic neural chips. From a co-design perspective, L²ight also provides essential insights for hardware-restricted unitary subspace optimization and efficient sparse training. We open-source our framework at link.

## 1 Introduction

The escalating scales of deep learning models and datasets have brought increased demand for computing capacities in electronic processors. Stringent performance and efficiency constraints in practical applications raise a surging need to develop more efficient computing solutions. As a promising substitute for conventional electronics, optical neural networks (ONNs) have attracted extensive research interests owing to their sub-nanosecond latency and attojoule/multiply-accumulate operation (MAC) energy efficiency [42, 6, 51, 41, 54, 14, 15], shown in Figure 1(a).

However, robustness and trainability are still critical issues for photonic AI engines [58, 22, 60]. Due to the analog computing nature of ONNs, the photonic DNN model inevitably suffer from performance degradation or even complete malfunction [58, 60] with the existence of manufacturing errors, non-ideal device controls, and undesired circuit noises, shown in Figure 1(b). Though non-ideal effects can be simulated and considered during software training [58, 22] to improve noise tolerance, the variation simulation is physically inaccurate (especially with unknown process variations) and prohibitively expensive, shown in Figure 1(c).

35th Conference on Neural Information Processing Systems (NeurIPS 2021).

Recently, on-device training has become an appealing trend towards adaptable and self-learning ONNs. However, training on photonic neural chips is non-trivial and much less explored than on conventional platforms. Prior work [57, 25, 21, 18] only demonstrated small prototypes, and their scalability and efficiency are rather

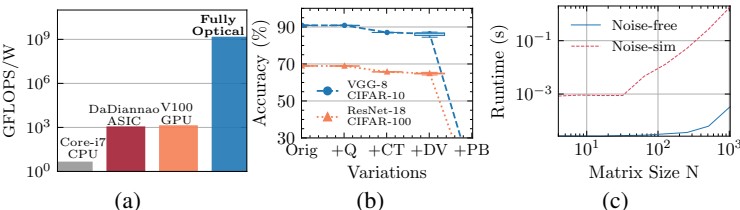

Figure 1: Comprehensive motivations. (a) Computational efficiency superiority of ONNs [42]. (b) Noise sensitivity of ONNs (Q: 8-bit quantization, CT: crosstalk, DV: device variation, PB: phase bias). (c) Runtime of noise-free matrix multiplication vs. w/ noise simulation (Q+CT+DV).

limited. To push the limits of DNNs in optics, we propose an efficient three-stage learning framework L²ight that consists of variation-agnostic identity calibration, alternate projection-based parallel mapping, and multi-level sparse subspace learning. The main contributions of this work are four-fold,

- **Scalability**. *For the first time*, an ONN learning protocol can scale up to million-level parameters under practical circuit non-ideality, over 3-order-of-magnitude more scalable than prior arts.

- **Efficiency**. We explore multi-level sparsity in *in-situ* gradient evaluation to trim down unnecessary on-chip training energy and runtime cost.

- **Learnability**. By trading redundant representability, our restricted subspace optimization can provide ONNs with enough adaptability for on-device self-learning and task transfer.

- **Robustness**. Various practical device noises and process variations are considered *in situ* to facilitate noise-resilient photonic AI engines.

- To our best knowledge, this is the first framework that supports on-chip training on million-parameter ONNs, over **1000×** more scalable and **30×** more efficient than prior art. We open-source a PyTorch-centric [38] ONN library torchonn and release our on-chip training framework at link.

## 2   Related Work

**Optical Neural Network and Training Methods.** One of recent ONN architectures adopts singular value decomposition (SVD) to implement matrix multiplication [42], i.e., $y = Wx = U\Sigma V^*x$. Cascaded 2-by-2 optical devices, i.e., Mach-Zehnder interferometers (MZIs), are used to construct unitary matrices as the product of a series of 2-dimensional unitary rotators, $U(n) = D \prod_{i=n}^{2} \prod_{j=1}^{i-1} R_{ij}(\phi_{ij})$. A detailed introduction to ONNs can be found in Appendix A. Beyond offline training [22], ONN on-chip training methods are proposed to offload the process back onto photonics [25, 21, 18], shown in Table 1. Brute-force device tuning (BFT) [42, 59] and evolutionary algorithms [57] are applied to search MZI settings. An adjoint variable method (AVM) [25] is proposed to directly evaluate gradients using *in-situ* light field monitoring. Stochastic zeroth-order optimization (ZOO) [21, 18] is later applied to improve the training efficiency. However, prior methods are hard to scale to larger ONNs either due to algorithmic inefficiency or unrealistic hardware complexity.

Table 1: Scalability comparison with prior ONN on-chip training protocols in terms of #Params they can handle, used algorithm, resolution requirement (*Req.*), and circuit observability requirement. *Coh. I/O* is short for coherent input/output [33, 56]. ZO, FO mean zeroth- and first-order methods.

|  | BFT [42] | PSO [57] | AVM [25] | FLOPS [21] | MixedTrn [4] | L²ight |
|---|---|---|---|---|---|---|
| #Params | ∼100 | ∼100 | ∼100 | ∼1000 | ∼2500 | **∼10 M** |
| Algorithm | ZO | ZO | FO | ZO | ZO | ZO+FO |
| Resolution Req. | Medium | High | Medium | High | Med | Medium |
| Observability Req. | Coh. I/O | Coh. I/O | Coh. I/O + Per device monitor | Coh. I/O | Coh. I/O | Coh. I/O |

**Efficient NN Training Methods.** Extensive work has been devoted to accelerating DNN training, among which an important branch is sparse backpropagation. Previous methods mainly focus on approximating matrix multiplication by sparsifying the pre-activation gradients [44], forward and feedback matrices [1, 39], and input feature maps [37]. Quantization to the pre-activation gradients is

adopted in [52] to induce sparsity by trading off quantization steps and performance. Other methods also exist, e.g., distributed and low-precision training [2, 3, 26]. However, they are not readily applicable to analog photonic engines, thus not in the scope of our discussion.

**Subspace Neural Networks.** Subspace neural networks are special DNN models with restricted parameter space but demonstrate comparable representability to classical NNs. Sparse NNs [23, 50], low-rank NNs [9, 28, 45], structured NNs [11, 30, 48], Fourier-domain NNs [19, 35, 34], and general frequency-domain NNs [20] were introduced to trim down the redundancy in DNNs by restricting the NN structure, matrix rank, numerical resolution, etc. In this work, we deeply explore the trade-off between ONN learnability, trainability, and efficiency in the restricted unitary subspace.

**Challenges of ONN On-Chip Training.** As a unique hardware-restricted optimization problem, ONN *in-situ* learning encounters fundamental challenges causing scalability issues in prior methods:

- **Lack of full-observability for *in-situ* light field.** Tracking physical optical field on every waveguide in $U$ and $V^*$ is not scalable or practical when ONNs scale up. Per device light field monitoring and calibration [17, 25] involves intractable hardware complexity. In practice, only $\Sigma$ can be precisely monitored and efficiently tuned.

- **Limited input/output observability**. In photonic tensor cores, for efficiency consideration, only the final output signals after $U\Sigma V^*$ can be coherently detected. Intermediate signals of a single unitary projection can not be easily read out without extra hardware support.

- **Inaccessible gradients for most control variables.** Due to the above two limitations, it is challenging to obtain true derivatives w.r.t. the MZI rotation phases in $U$ and $V^*$ [42, 21, 18], casting fundamental *in-situ* optimization difficulty as ONN scales up.

To enable *in-situ* self-learning for ONNs, the proposed synergistic framework $\texttt{L}^2\texttt{ight}$ provides a scalable, efficient, and on-chip-implementable solution that overcomes those hardware restrictions.

## 3 Synergistic ONN On-Chip Learning Framework $\texttt{L}^2\texttt{ight}$

In this section, we give a formal description of the ONN on-chip training problem and detailed demonstration of our proposed three-stage learning flow $\texttt{L}^2\texttt{ight}$, shown in Figure. 2.

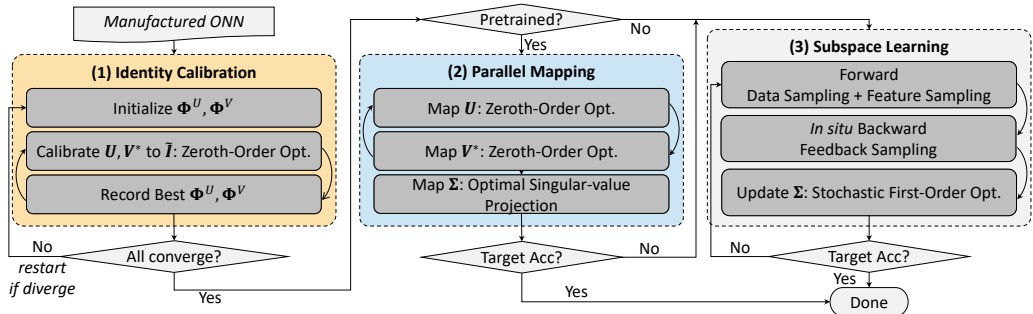

Figure 2: Proposed three-stage ONN on-chip learning flow $\texttt{L}^2\texttt{ight}$.

### 3.1 Understanding the ONN On-Chip Learning Problem

The ONN that supports on-chip learning is shown in Figure 3, constructed by local storage, control units, interconnects, and photonic tensor cores with coherent I/O [33, 56] and wavelength-division multiplexing (WDM) [55, 46] for parallel processing. The target is to optimize MZI phases $\Phi$ directly on chip under variations. Formally the *hardware-restricted* learning problem is,

$$\Phi^* = \operatorname*{argmin}_{\Phi} \ \mathcal{L}\big(W(\Omega\Gamma\mathcal{Q}(\Phi) + \Phi_b); \mathcal{D}_{trn}\big),$$

$$\text{s.t. } W(\Phi) = \big\{W_{pq}(\Phi_{pq})\big\}_{p=0,q=0}^{p=P-1,q=Q-1}, \quad W_{pq}(\Phi_{pq}) = U_{pq}(\Phi_{pq}^U)\Sigma_{pq}(\Phi_{pq}^S)V_{pq}^*(\Phi_{pq}^V),$$

$$U_{pq}(\Phi_{pq}^U) = D_{pq}^U \prod_{i=k}^{2} \prod_{j=1}^{i-1} R_{pqij}(\phi_{pqij}^U), \quad V_{pq}^*(\Phi_{pq}^V) = D_{pq}^V \prod_{i=k}^{2} \prod_{j=1}^{i-1} R_{pqij}(\phi_{pqij}^V), \tag{1}$$

$$\Sigma_{pq}(\Phi_{pq}^S) = \max(|\Sigma_{pq}|)\texttt{diag}(\cdots, \cos\phi_{pq,i}^S, \cdots), \quad \Phi_b \sim \mathcal{U}(0, 2\pi), \ \Gamma \sim \mathcal{N}(\gamma, \sigma_\gamma^2).$$

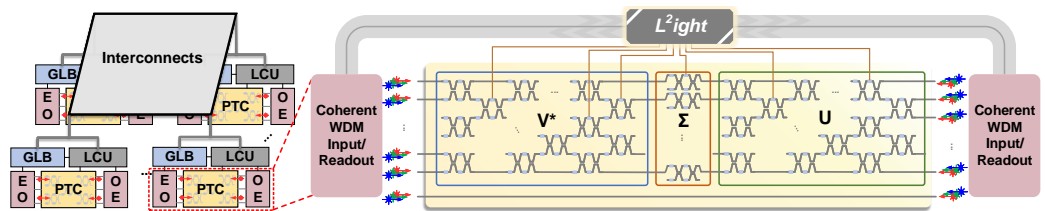

Figure 3: ONN architecture. PTC: photonic tensor core, GLB: global buffer, LCU: local control unit, EO: electrical-to-optical conversion.

The linear projection in an ONN adopts blocking matrix multiplication, where the $M \times N$ weight matrix is partitioned into $P \times Q$ blocks of size $k \times k$. During the optimization of $\Phi$, we jointly consider control resolution limits $\mathcal{Q}(\cdot)$ [22, 42], device process variations $\Gamma$ [22, 21, 18], thermal crosstalk among adjacent devices $\Omega$ [21, 60], and unknown phase bias due to manufacturing error $\Phi_b$ for *in-situ* robustness-aware training. A detailed non-ideality analysis is in Appendix A.3. For practicality, robustness, and convergence consideration, we select $k=9$, which is explained in Appendix F.

### 3.2 Identity Calibration (IC): Variation-Agnostic Circuit State Preparation

After manufacturing, unknown process variations in waveguides make the initial state of PTCs unpredictable [47, 60]. A primary task is to prepare $U$ and $V^*$ to be identity matrices. However, the calibration problem, i.e., $\min_{\Phi^U, \Phi^V} \sum_{p,q} \left( \|U_{pq}(\Phi_{pq}^U) - I\|_2^2 + \|V_{pq}^*(\Phi_{pq}^V) - I\|_2^2 \right)$, is not solvable given the observability and controllability constraints on $U$ and $V^*$. The closest auxiliary problem that we can solve is the one with absolute operations on unitaries, i.e., $\min_{\Phi^U, \Phi^V} \sum_{p,q} \left( \||U_{pq}(\Phi_{pq}^U)| - I\|_2^2 + \||V_{pq}^*(\Phi_{pq}^V)| - I\|_2^2 \right)$. We denote those two mean square errors as $MSE^U$ and $MSE^V$. We rewrite it as a surrogate minimization of $\mathcal{L}_{IC}$ that can lead to the same solution,

$$\min_{\Phi} \sum_{p,q} \|U_{pq}(\Phi_{pq}^U) \Sigma_{pq} V_{pq}^*(\Phi_{pq}^V) \Sigma_{pq}^{-1} - I\|. \quad (2)$$

The optimal solution for this auxiliary problem is $U = V^* = \tilde{I}$, where $\tilde{I}$ is not guaranteed to be an identity matrix but a more general *sign-flipping matrix* with *arbitrary and unobservable sign flips* on the same columns in $U$ and rows in $V^*$, shown in Figure 4(a). We adopt zeroth-order optimization (ZOO) on $\Phi^U$ and $\Phi^V$ to calibrate $U$ and $V^*$ to approach $\tilde{I}$, shown in Figure 4(b). We show the converged solution of Eq. (2) with unobservable sign flips and suboptimality only has marginal impacts on the following training procedure in later sections.

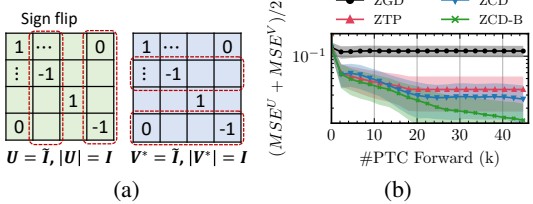

Figure 4: (a) Identity calibration with sign flip. (b) Different ZO optimizers on identity calibration. (ZGD: ZO gradient descent with momentum, ZCD: ZO coordinate descent, ZTP: ZO three-point. *B* is best solution recording.)

### 3.3 Parallel Mapping (PM): Alternate Projection-based Model Deployment

The target is to map the pre-trained weights $W$ onto photonic MZI meshes $\widetilde{W}(\Phi)$ with high fidelity. We formulate the parallel mapping as a batched $k \times k$-blockwise regression problem,

$$\min_{\Phi} \sum_{p,q} \|\widetilde{W}_{pq}(\Phi_{pq}) - W_{pq}\|_2^2. \quad (3)$$

As analyzed before, $\frac{\partial W}{\partial \Phi^U}$ and $\frac{\partial W}{\partial \Phi^V}$ are too expensive to compute *in situ*. We propose a parallel mapping flow with alternate zeroth-order optimization on $\Phi^U$ and $\Phi^V$. After convergence, we will perform analytical optimal singular-value projection (OSP) to minimize the regression error given fixed singular vectors.

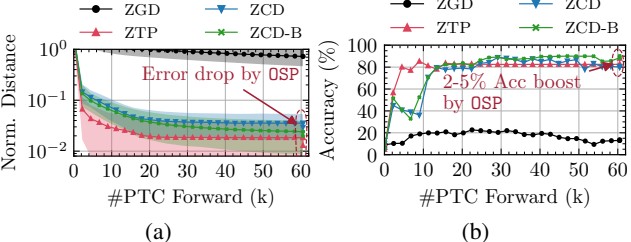

Figure 5: ZTP and ZCD-B perform the best in parallel mapping. The optimal singular-value projection leads to significant error drop and accuracy jump.)

We show why OSP gives the optimal solution under sign flips and how to perform it on the PTC.

**Claim 1.** *Optimal singular-value projection (OSP): the optimal singular value problem, i.e., $\Sigma_{opt} = \text{argmin}_\Sigma \|U\Sigma V^* - W\|$, can be analytically solved on-chip with arbitrary and unknown sign flip.*
*Proof.*

$$\Sigma_{opt} = \text{diag}\big(U^{-1}W(V^*)^{-1}\big) = \text{diag}\big(U^*WV\big) = \text{diag}\big((\tilde{I}^*V^*W^*U\tilde{I})^*\big). \quad (4)$$

OSP can be directly achieved using the limited operation set, i.e., $\{U, U^*, V, V^*\}$, supported by the reciprocal PTC itself. Specifically, we configure $V^* = \tilde{I}$ and $\Sigma = I$, and shine in a coherent WDM light beam that carries $W$ from right ports. Since the coherent photonic circuit is reciprocal [33], we can read $\tilde{I}U^*W$ on the left ports. Then we configure $U = \tilde{I}$ and $\Sigma = I$, and shine in its adjoint field from left, i.e., $W^*U\tilde{I}^*$. We can directly read out the projected optimal diagonal on the right because the sign flips in the unitary matrices naturally cancel out on the diagonal. □

Figure 5 compares different ZO optimizers on this task. Coordinate-wise optimizers (ZCD [31] and ZTP [13]) outperform the gradient-based ZGD [16] with higher accuracy and convergence speed. This procedure is highly parallel and efficient since the mapping involves *no stochasticity* and only happens *locally* within each PTC. We can also observe that OSP effectively reduces the normalized matrix distance ($\|W - \widetilde{W}\|_2^2 / \|W\|_2^2$) and boosts the accuracy by 2-5% almost for free.

### 3.4 Subspace Learning: Hardware-Aware Multi-Level Sparse Training

Besides mapping from an offline-trained model, L²ight also supports *in-situ* self-learning fully on chip. We name this feature as *subspace learning*. To make L²ight hardware-aware, we trade expensive full-space trainability for efficient subspace gradient evaluation, i.e., $\frac{\partial\mathcal{L}}{\partial\Sigma}$ which coincides with the general frequency-domain ONNs [19, 20] and subspace NN design concept [43]. Since this learning stage involves stochasticity, it turns out to be the efficiency bottleneck, especially the backward pass. Hence, we explore multi-level sparsity for efficient *in-situ* gradient approximation.

#### 3.4.1 *In-situ* Subspace Gradient Acquisition via Reciprocity in Optics

The conventional way to compute first-order gradients w.r.t. $\Sigma$ is $\frac{\partial\mathcal{L}}{\partial\Sigma} = \text{diag}\big(U^*\frac{\partial\mathcal{L}}{\partial W}V\big)$. However, $\frac{\partial\mathcal{L}}{\partial W} = \frac{\partial\mathcal{L}}{\partial y}x^T$ requires arbitrary matrix multiplication, which is not implementable by weight-stationary PTCs. Hence, we remap it as,

$$\frac{\partial\mathcal{L}}{\partial\Sigma} = \frac{\partial\mathcal{L}}{\partial y_\Sigma} \odot y_V = \big(\tilde{I}U^*\frac{\partial\mathcal{L}}{\partial y}\big) \odot (\tilde{I}V^*x). \quad (5)$$

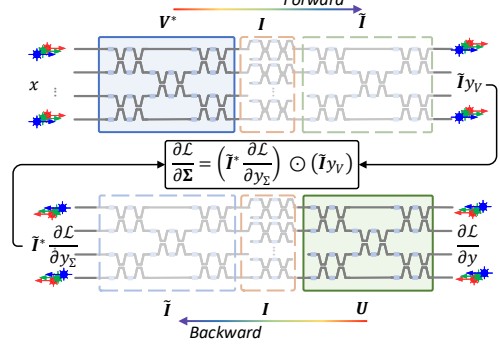

By shining in coherent WDM beams carrying the inputs and upstream gradients forward and backward through the reciprocal PTCs, respectively, as shown in Figure 6, the weight gradients can be efficiently obtained with lightweight element-wise multiplication $\odot$, which can be offloaded to electrical units. $\tilde{I}$ naturally cancels out by the Hadamard product with no impacts on gradient fidelity.

Figure 6: *In-situ* subspace gradient acquisition.

#### 3.4.2 Multi-Level Sparse Subspace Learning

Inspired by sparse backpropagation methods [44, 37, 49, 36, 39], we propose multi-level sparse subspace learning to cut down both energy cost and total time steps in on-chip gradient evaluation.

**Balanced Feedback Sampling.** To improve the efficiency of the error feedback process, i.e., $W^T\frac{\partial\mathcal{L}}{\partial y}$, as shown in Figure 7, we sample the feedback matrix $W^T \in \mathbb{R}^{N \times M}$ with a structured sparse mask $\mathcal{P}_W = c_W(\mathcal{S}_W \otimes \mathbf{1})$ generated by the Kronecker product between a boolean mask $\mathcal{S}_W \in \{0,1\}^{Q \times P}$ with sparsity $\alpha_W$ and an all-ones matrix $\mathbf{1}$, where the scaling factor $c_W$ is set to $\frac{1}{\alpha_W} = \frac{PQ}{\text{Tr}(\mathcal{S}_W^T \mathcal{S}_W)}$ for unbiased estimation, proven in Appendix D. The efficiency benefits come from two aspects: (1) the structurally masked PTCs are entirely idle, directly saving energy, and (2) the product accumulation depth/step is reduced by a factor of $\alpha_W$, effectively trimming time steps.

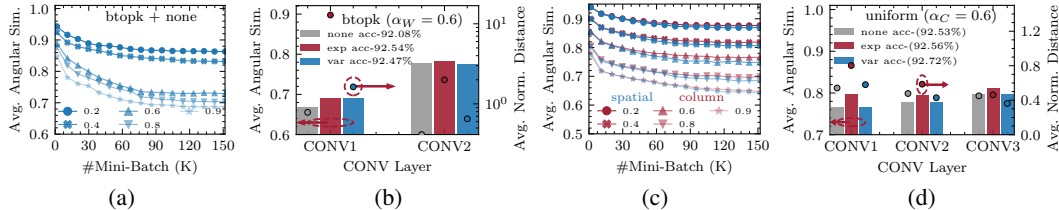

(a)  (b)  (c)  (d)

Figure 8: Average gradient angular similarity with different feedback sparsity (a) and three normalization methods (b). *none, exp,* and *var* represents no, expectation-maintained, and variance-maintained normalization. Average gradient angular similarity with spatial and column sampling (c) and three normalization methods (d).

However, two major downsides exist on traditional `uniform` and layer-wise `topk` sampling [39]. First, on a backward path, multiple feedback sampling operators will be cascaded, such that importance-unaware `uniform` sampling can lead to an exponentially large variance [37]. Second, `topk` sampling is overly greedy and tends to break the load balance as the feedback latency can be bottlenecked by the longest partial product accumulation path, shown in Figure 7. To tackle this, we propose a balanced top-K sampling (`btopk`) to draw $\mathcal{S}_W$ from a guided distribution that locally prefers blocks with large Frobenius norm, which can be efficiently evaluated by $\|\boldsymbol{W}_{pq}\|_{\mathcal{F}}^2 = \mathrm{Tr}(|\boldsymbol{\Sigma}_{pq}|^2)$. It strikes a *balance between gradient variance and bias* by fine-grained row-wise top-K sampling and *eliminates load-imbalance* by guaranteeing the same sparsity for different rows of $\boldsymbol{W}^T$, i.e., $\sum_p \mathcal{S}_W(1,:) = \sum_p \mathcal{S}_W(2,:) = \cdots = \sum_p \mathcal{S}_W(Q,:)$. Figure 8(a), 8(b) shows the gradient approximation fidelity in terms of average angular similarity [5] and normalized matrix

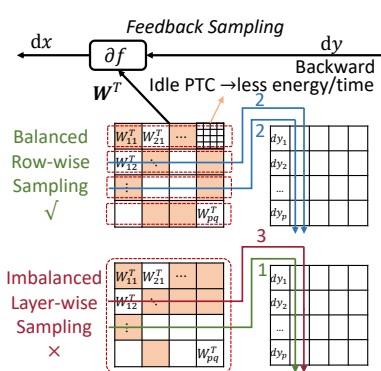

Figure 7: Balanced v.s. imbalanced feedback matrix sampling.

distance. Our `btopk`-sampled weight gradients align well with the true gradients. With the unbiased (`exp`) normalization factor $\alpha_W$, `btopk` shows the best gradient angular similarity and inference accuracy compared with others.

**Information-Preserving Column Sampling.** Input feature sparsification can also effectively cut down the gradient evaluation cost [39, 37], especially for costly CONV layers. However, with traditional *spatial sampling* (SS) [39, 37], the input feature map $x$ barely maintains its sparsity regularity after being transformed to flattened patches $X$ via *im2col* if the kernel size is larger than 1, shown in Figure 9. Hence, we propose a novel *column sampling* (CS) as a better solution. We sample $X$ using a mask $\mathcal{S}_C\{0,1\}^{H'W'}$ with a uniform sparsity $\alpha_C$, which is shared across batches with negligible overhead. This leads to both information preservation and efficiency improvement. First, in Figure 9, a pixel appears in multiple columns, such that partial information can be maintained after column sampling. Second, this highly-structured column dropping directly translates to less PTC forward energy and fewer partial gradient accumulation steps. In contrast, with a spatial mask $\mathcal{S}_S$ and spatial sparsity $\alpha_S$, the masked pixel will be completely dropped with poor regularity after *im2col*, at the cost of large variance due to information loss and almost no runtime improvement on this dense linear projection engines. Note that for CONV1×1, CS turns out to be equivalent to SS, which can simultaneously save memory and runtime. Figures 8(c), 8(d) show that our proposed

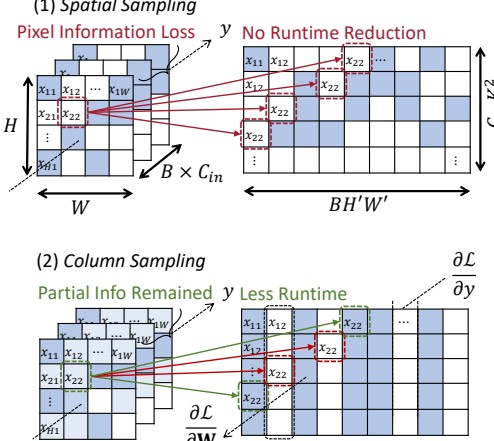

Figure 9: Spatial and column sampling for CONV.

CS can obtain better gradient approximation fidelity than prior SS. Different normalization has small effects on model accuracy since feature sampling only happens locally within each layer, without any variance cascade effect. Note that simultaneous scaling by $\alpha_W$ and $\alpha_C$ tends to generate overly-

confident gradient approximation, which empirically leads to harmful gradient variance. Hence, we will adopt $\alpha_C$=1 in all experiments.

**Data Sampling.** After parallel mapping, the ONN is initialized fairly close to the target pre-trained model. It is reasonable and intuitive to calibrate it with a representative calibration set instead of the entire training set. Inspired by the mini-batch dropping (SMD) technique [49], we integrate this SMD technique into our framework to further explore data-level sparsity. Within one training epoch, we randomly skip each iteration with probability $\alpha_D$, directly translating to training time reduction.

### 3.5 Complexity Analysis of Three Stages in L²ight

We assume the total step in IC, PM, and SL is $T_1$, $T_2$, and $T_3$, respectively. The ONN has $L$ layers, each including an $N \times N$ weight matrix partitioned into multiple $k \times k$ blocks.

**Identity Calibration and Parallel Mapping.** Each block optimizes $k(k-1)$ phases using ZOO. All $LN^2/k^2$ blocks are optimized in parallel. The total step is $2k(k-1)T_1$ for IC and $2LN^2(k-1)T_2/k + 3$ for PM. The total PTC call is around $2LN^2T_1$ or $2LN^2T_2$ for IC and PM, respectively.

**Subspace Learning.** We assume the feature map size is $H \times W$ with a batch size of $B$. The detailed complexity analysis is given in Appendix G. The total step is approximately $T_3LNBHW/k$.

According to our training cost profiler, IC and PM in total is 3-order-of-magnitude cheaper than the SL stage, since the batched parallel regression is deterministic and data-independent.

## 4 Results

### 4.1 Experiment Setup

**Datasets.** We evaluate L²ight on Vowel [10], MNIST [29], FashionMNIST [53], CIFAR-10, CIFAR-100 [27], and TinyImagenet [7]. On CIFAR-10/100 and TinyImagenet, we adopt random crop, flip, color jittering for augmentation.

**Models.** All models are implemented with our open-source PyTorch-centric ONN library torchonn. We evaluate on a customized MLP (8-16-16-4) [18] on Vowel, CNN-S (CONV8K3S2-CONV6K3S2-FC10) [18] on MNIST, a CNN-L ({CONV64K3}×3-Pool5-FC10) on FashionMNIST, and VGG-8 [8]/ResNet-18 [1] [24] on CIFAR-10/100. CNN-L/FashionMNIST is used for ablation studies. VGG-8/ResNet-18 on CIFAR-10/100 are used for accuracy and efficiency comparison. Training details can be found in Appendix E.

**Efficiency Evaluation.** We assume fully parallel $9 \times 9$-blocking matrix multiplication in photonic tensor cores and sequential partial product accumulation in electronics. All experiments and performance measurements are based on software simulation with various noise modeling. Our simulator counts the total number of PTC calls as the normalized energy indicator and the longest accumulation path as the normalized latency/runtime indicator. Details of profiling can be found in Appendix G.

### 4.2 Main Results

**Scalability Comparison with Prior ONN Learning Protocols.** Figure 10 compares L²ight with two SOTA ONN on-chip training protocols, FLOPS [21] and MixedTrn [18]. For ZO methods, i.e., FLOPS and MixedTrn, we count the energy and latency of forward PTC query in Appendix G. Prior protocols can only handle toy models ($\sim$2,000 params) given their algorithmic inefficiency and instability, while our L²ight shows >1,000× higher scalability to handle large ONNs ($\sim$10 M) on challenging tasks with comparable accuracy to full-space pre-trained models. Though MixedTrn achieves com-

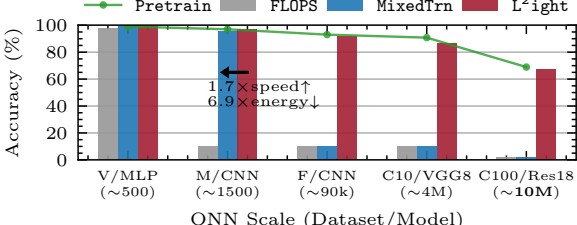

Figure 10: Compare scalability with prior protocols [21, 18].

[1] https://github.com/kuangliu/pytorch-cifar

parable accuracy to L$^2$ight on small benchmarks, we are still 1.7× faster and 6.9× more energy efficient.

The superiority of L$^2$ight provides three important insights: (1) *decoupling ZOO from stochasticity* and *partitioning a large-scale regression problem into a batch of sub-tasks* can greatly mitigate the curse of dimensionality both in convergence and efficiency. (2) *mapping before learning* can fully leverage the pre-trained model to reduce the learning cost. Prior methods have to learn from a severely corrupted solution under variations, while L$^2$ight recovers most accuracy via mapping, leaving a very light workload for subspace learning. (3) Restricted subspace learning provides *adequate degree of freedom* for training from scratch and task transfer. Also, its *compatibility with first-order* methods significantly boosts the trainability and breaks the scalability barrier for ONN training. We now validate the above insights by extensive experiments.

**Training Efficiency Comparison with Prior Sparse Training Methods.** In Figure 11, we show accuracy and efficiency comparison of 1) baseline L$^2$ight-SL (BS), 2) L$^2$ight-SL with spatial sampling (RAD), 3) L$^2$ight-SL with weight and spatial sampling (SWAT-U), and 4) L$^2$ight-SL with all three introduced sampling methods (feedback, column, and data sampling), and 5) our proposed full flow with IC, PM, and sparse SL (L$^2$ight). To clarify, L$^2$ight-SL performs subspace learning on-chip from scratch without using pre-trained weights, while L$^2$ight includes the full flow, i.e., pre-training, mapping, and on-chip training. When we perform subspace learning from scratch, our proposed *multi-level sampling* strategies outperform previous RAD and SWAT-U by ∼**3×** in hardware cost with comparable accuracy. Though RAD can save the forward peak memory, it leaves the most expensive backward pass unoptimized, which does not fully exploit the sparsity in ONN training. SWAT-U tries to save forward cost by simultaneously sparsifying the forward and feedback weights with shared masks/patterns. However, in our experiment, the forward sparsification considerably degrades the performance, which dilates the efficiency benefits from it. Parallel mapping can fully leverage the pre-trained weights and help our full three-stage flow L$^2$ight achieve the best accuracy with *much faster convergence*, leading to **over 30×** higher energy efficiency and fewer time steps.

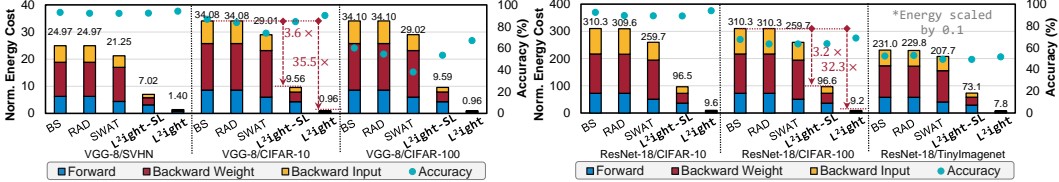

Figure 11: Accuracy and hardware efficiency comparison on VGG-8 (*left*) and ResNet-18 (*right*).

Note that the energy efficiency and latency improvement is *not just on the photonic part but a systematic performance boost*. Our three-level sampling methods directly skip the pruned block, which means the corresponding cost of memory transaction, computation, control, and communication are removed together. Therefore, the sampling sparsity can be directly translated to the energy/latency improvement ratio regardless of whether the electrical part dominates the total cost.

## 4.3 Ablation Studies and Discussion

### 4.3.1 Multi-Level Sparsity in Efficient Training

**Feedback Sparsity.** To investigate the impact of feedback sampling strategies, we visualize the gradient approximation fidelity and accuracy curves in Figure 12(a). uniform sampling shows varying performance under different sparsity values due to large gradient variances. topk shows worse performance after sufficient steps due to its biased gradient estimation from overly greedy block selection. In contrast, our proposed *load-balancing* btopk strikes a balance between variance and bias via block-wise sampling and also leads to less runtime as it forces load balance among massively parallel PTCs. In Table 2, feedback sampling saves 50-60% time steps on the most costly error feedback $\nabla_x\mathcal{L}$, leading to 1.5-1.8× overall time step reduction with minimum accuracy drop.

**Feature Sparsity.** Figure 12(b) compares the accuracy and weight gradient computation time steps on two feature sampling techniques. Though *spatial sampling* (ss) can save peak storage by dropping a subset of activations during the forward pass, it shows no gradient computation step reduction. Our hardware-friendly *column sampling* (cs) directly leads to energy and runtime reduction due to its

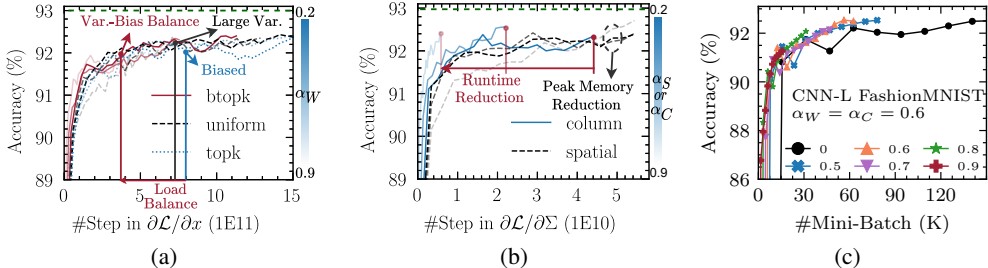

(a)           (b)           (c)

Figure 12: Accuracy v.s. weight gradient computation steps with three feedback sampling strategies (a) and different feature sampling techniques (b). Accuracy (93.02%) from a full-space trained model (green). CNN-L/FashionMNIST is used for (a) and (b). Compare different data sampling sparsity (c).

structured sparsity. In Table 2, when column sampling is further added, we observe ∼50% PTC energy saving on weight gradient computation $\nabla_\Sigma \mathcal{L}$ at the cost of ∼1% accuracy drop.

**Data Sparsity.** In the data level, we also demonstrate how SMD with sparsity $\alpha_D$ impacts the training efficiency in Figure 12(c). With the best selected $\alpha_W$ and $\alpha_C$, data sparsity directly reduces training time by skipping iterations [49]. The data sampling selects a uniform subset of the training set to represent the original data distribution, leading to less data processing with comparable generalization in the extracted features. Another explanation is that the variance increased by partial replacement serves as a regularization mechanism to improve the model performance [49]. For relatively easy tasks, aggressive sparsity ($\alpha_D$=0.8) is a sweet point, while for larger datasets shown in Table 2, a medium sparsity (0.5) can be a good setting to balance both the training cost and accuracy. With all three sampling methods integrated, our L²ight-SL shows competitive accuracy and ∼3× higher efficiency than RAD and SWAT-U. More advanced dataset sampling methods are left for future exploration.

### 4.3.2 Learnability of Restricted Subspace ONNs

**Impacts of Calibration/Mapping Quality.** Table 2 shows that with IC and PM, the full L²ight flow achieves the highest accuracy with 32-35× efficiency boost over baselines. We further evaluate the impact of different mapping accuracy and the calibration quality on subspace learning in Figure 13. First, *parallel mapping or pre-training is not a must*. Our subspace learning supports first-order optimization on-chip from random initialization. Second, *the optimality on subspace bases influences the final accuracy* as it determines the upper bound of accuracy that can be recovered by subspace learning. With roughly optimized space bases, i.e., $U, V^*$, subspace learning can efficiently train basis coefficients, i.e., $\Sigma$, achieving 5-6% higher accuracy and 9.9× less energy and steps compared with random unitaries (train from scratch). Third,

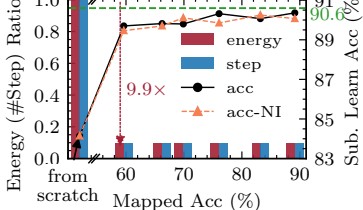

Figure 13: Impact of mapping accuracy (VGG-8 CIFAR-10 with $\alpha_W$=$\alpha_C$=0.6, $\alpha_D$=0.5). *acc-NI* is the curve with non-ideal $\tilde{I}$.

*subspace optimization shows low sensitivity on mapping quality* and is able to compensate for the suboptimality in singular vectors within a reasonable range. Even with 60% mapped accuracy,

Table 2: Compare sampling strategies on CIFAR-10 in terms of accuracy, activation size reduction, energy, and time step. Forward, weight gradient, and error feedback are denoted as $\mathcal{L}$, $\nabla_\Sigma \mathcal{L}$, and $\nabla_x \mathcal{L}$. L²ight-SL is learning *from scratch*, and L²ight (IC→PM→SL) is the full flow with pre-trained weights and non-ideal $\tilde{I}$.

| | Acc$_{\pm\sigma}$ (%) | Act↓(%) | Norm. PTC Energy | | | | Norm. #Step | | | |
|---|---|---|---|---|---|---|---|---|---|---|
| | | | $\mathcal{L}$ | $\nabla_\Sigma\mathcal{L}$ | $\nabla_x\mathcal{L}$ | Total (Ratio) | $\mathcal{L}$ | $\nabla_\Sigma\mathcal{L}$ | $\nabla_x\mathcal{L}$ | Total (Ratio) |
| L²ight-SL (Baseline) VGG-8 | 86.66$_{\pm0.13}$ | - | 8.58 | 17.16 | 8.34 | 34.08 (1.00) | 32.64 | 5.49 | 92.02 | 130.14 (1.00) |
| + Feedback Sampling ($\alpha_W$=0.6) | 86.41$_{\pm0.25}$ | - | 8.58 | 17.16 | 3.38 | 29.13 (1.17) | 32.64 | 5.49 | 35.76 | 73.89 (1.76) |
| + Column Sampling ($\alpha_C$=0.6) | 85.58$_{\pm0.01}$ | - | 8.58 | 7.16 | 3.38 | 19.12 (1.78) | 32.64 | 4.67 | 35.76 | 73.07 (1.78) |
| + Data Sampling ($\alpha_D$=0.5) | 84.45$_{\pm0.45}$ | - | 4.29 | 3.58 | 1.69 | 9.56 (3.56) | 16.32 | 2.34 | 17.89 | 36.54 (3.56) |
| + RAD [37] ($\alpha_S$=0.85) | 83.68$_{\pm0.58}$ | 11.78 | 8.58 | 17.16 | 8.34 | 34.08 (1.00) | 32.64 | 5.49 | 92.02 | 130.14 (1.00) |
| + SWAT-U [39] ($\alpha_W$=0.3, $\alpha_S$=0.6) | 73.91$_{\pm0.27}$ | 8.31 | 6.01 | 17.16 | 5.84 | 29.01 (1.17) | 25.98 | 5.49 | 82.19 | 113.66 (1.15) |
| L²ight (IC→PM→SL) | **90.20$_{\pm0.05}$** | - | **0.43** | **0.36** | **0.17** | **0.96 (35.64)** | **1.63** | **0.23** | **1.79** | **3.65 (35.64)** |
| L²ight-SL (Baseline) ResNet-18 | 92.37$_{\pm0.08}$ | - | 72.24 | 144.49 | 93.60 | 310.33 (1.00) | 463.40 | 27.23 | 1,478.84 | 1,969.48 (1.00) |
| + Feedback Sampling ($\alpha_W$=0.5) | 91.35$_{\pm0.03}$ | - | 72.24 | 144.49 | 48.13 | 264.86 (1.17) | 463.40 | 27.23 | 747.22 | 1,237.85 (1.59) |
| + Column Sampling ($\alpha_C$=0.5) | 90.02$_{\pm0.16}$ | 4.47 | 72.24 | 72.49 | 48.13 | 192.86 (1.61) | 463.40 | 15.68 | 747.21 | 1,226.30 (1.61) |
| + Data Sampling ($\alpha_D$=0.5) | 89.07$_{\pm0.04}$ | 4.47 | 36.13 | 36.26 | 24.07 | 96.46 (3.22) | 231.76 | 7.84 | 373.71 | 613.31 (3.21) |
| + RAD [37] ($\alpha_S$=0.9) | 89.44$_{\pm0.17}$ | 46.60 | 72.26 | 143.72 | 93.60 | 309.58 (1.00) | 463.53 | 26.03 | 1,478.84 | 1,969.00 (1.00) |
| + SWAT-U [39] ($\alpha_W$=0.3, $\alpha_S$=0.5) | 89.21$_{\pm0.16}$ | 25.89 | 50.57 | 143.64 | 65.52 | 259.73 (1.19) | 358.40 | 26.56 | 1,417.96 | 1,802.00 (1.09) |
| L²ight (IC→PM→SL) | **93.91$_{\pm0.02}$** | 4.47 | **3.61** | **3.62** | **2.41** | **9.64 (32.20)** | **23.16** | **0.78** | **37.34** | **61.29 (32.13)** |

singular value optimization has enough capability to recover the accuracy to ~90%. Fourth, our subspace learning is *robust to gradient noises* caused by non-ideal $\tilde{I}$ ($MSE^U \approx MSE^V \approx 0.013$), which shows that L²ight can tolerate reasonable suboptimality in the calibration and mapping stages.

***In-situ* Transferability in the Restricted Subspace.** Another important question to answer is the

transferability of subspace learning. After mapping, we fix the inherited unitaries and adapt to different tasks by only training the singular values. Figure 14 shows that the inherited bases span a good design space with enough transferability. The *in-situ* subspace transfer learning shows 1-2% higher final accuracy. Also, it uses 3~5× fewer steps to obtain the same accuracy as training from scratch. Hence, our proposed L²ight finds a highly trainable design point while the learnability is still mostly maintained.

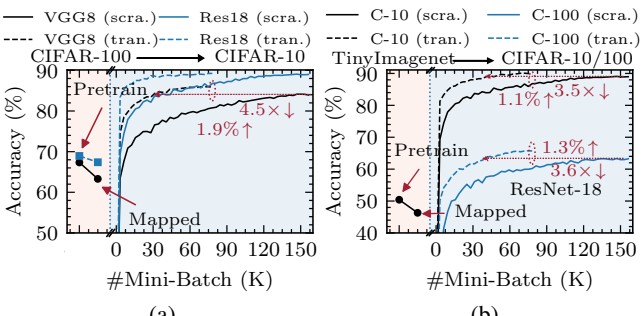

Figure 14: (a) Transfer VGG8/Res18 from CIFAR-100 to CIFAR-10. (b) Transfer Res18 from TinyImagenet to CIFAR-10 and 100.

## 5 Conclusion

In this work, we propose the *first* scalable and efficient on-chip learning framework L²ight for emerging optical neural networks. Our proposed three-stage flow synergistically enables on-chip self-learning via automatic circuit state calibration, parallel model mapping, and efficient subspace learning. To further improve the learning efficiency, we explore multi-level sparsity, including balanced feedback sampling, information-preserving column feature sampling, and runtime-reduced data sampling. Extensive ablation studies and comparison experiments show 3-order-of-magnitude scalability improvement over prior on-chip training protocols and 30× efficiency boost compared with previous sparse training methods. We open-source a PyTorch-centric ONN library torchonn, based on which we release our on-chip learning framework L²ight at link. In the future, we will go beyond current software simulation and experimentally validate the effectiveness of L²ight on real photonic neural chips.

**Acknowledgments** The authors acknowledge the Multidisciplinary University Research Initiative (MURI) program through the Air Force Office of Scientific Research (AFOSR), contract No. FA 9550-17-1-0071, monitored by Dr. Gernot S. Pomrenke.

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
