# A  ONN Principles

## A.1  Mach-Zehnder Interferometers (MZIs)

A basic coherent optical component used in this work is an MZI. One of the most general MZI structures is shown in Figure 15, consisting of two 50-by-50 optical directional couplers and four phase shifters $\theta_T$, $\theta_L$, $\omega_P$, and $\omega_W$. An MZI can achieve arbitrary 2×2 unitary matrices $SU(2)$. The

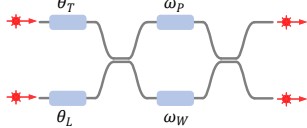

Figure 15: 2-by-2 MZI with top (T), left (L), upper (P), and lower (W) phase shifters.

physical transfer matrix $R(\theta_g, \Delta\theta, \Delta\omega)$ of an MZI shown in Fig. 15 is,

$$
\begin{aligned}
SU(2) = R(\theta_g, \Delta\theta, \Delta\omega) &= \begin{pmatrix} t & kj \\ kj & t \end{pmatrix} \begin{pmatrix} e^{j\omega_P} & 0 \\ 0 & e^{j\omega_W} \end{pmatrix} \begin{pmatrix} t & kj \\ kj & t \end{pmatrix} \begin{pmatrix} e^{j\theta_T} & 0 \\ 0 & e^{j\theta_L} \end{pmatrix} \\
&= e^{j\theta_g} \begin{pmatrix} \sin\frac{\Delta\omega}{2} & \cos\frac{\Delta\omega}{2} \\ \cos\frac{\Delta\omega}{2} & -\sin\frac{\Delta\omega}{2} \end{pmatrix} \begin{pmatrix} e^{j\frac{\Delta\theta}{2}} & 0 \\ 0 & e^{-j\frac{\Delta\theta}{2}} \end{pmatrix}, \\
\theta_g &= \bar\theta + \bar\omega + \frac{\pi}{2}, \ \bar\theta = \frac{\theta_T + \theta_L}{2}, \ \bar\omega = \frac{\omega_P + \omega_W}{2}, \\
\Delta\theta &= \theta_T - \theta_L, \ \Delta\omega = \omega_P - \omega_W, \ t = k = \frac{\sqrt{2}}{2}.
\end{aligned}
\tag{6}
$$

where the global phase $\theta_g$ is determined by the common mode $\bar\theta$ and $\bar\omega$, and the light splitting is determined by the differential mode $\Delta\theta$ and $\Delta\omega$. To achieve the 2-D planar rotator $R(2)$ in the real space parametrized by $\phi$, we let $\theta_T = \pi/2$, $\theta_L = 3\pi/2$, $\bar\omega = \pi$. To convert the simplified transfer matrix $M(\Delta\omega)$ to the planar rotator, we set $\Delta\omega = \pi - 2\phi$ as follows,

$$
\begin{aligned}
R(2) &= e^{j\frac{3\pi}{2}} \begin{pmatrix} \sin\frac{\Delta\omega}{2} & \cos\frac{\Delta\omega}{2} \\ \cos\frac{\Delta\omega}{2} & -\sin\frac{\Delta\omega}{2} \end{pmatrix} \begin{pmatrix} j & 0 \\ 0 & -j \end{pmatrix} \\
&= \begin{pmatrix} \sin\left(\frac{\pi-2\phi}{2}\right) & -\cos\left(\frac{\pi-2\phi}{2}\right) \\ \cos\left(\frac{\pi-2\phi}{2}\right) & \sin\left(\frac{\pi-2\phi}{2}\right) \end{pmatrix} = \begin{pmatrix} \cos\phi & -\sin\phi \\ \sin\phi & \cos\phi \end{pmatrix}.
\end{aligned}
\tag{7}
$$

## A.2  MZI-based Photonic Tensor Core Architecture

By cascading $N(N-1)/2$ MZIs into a triangular mesh (Recks-style) or rectangular mesh (Clements-style), we can construct arbitrary $N \times N$ unitary $U(N)$.

As a simple example, we show the principle of Recks-style MZI array for a simple demonstration. A similar decomposition can be derived for the Clements style. It decomposes an $M \times N$ weight matrix using SVD, i.e., $\boldsymbol{W} = \boldsymbol{U}\boldsymbol{\Sigma}\boldsymbol{V}^*$. The diagonal matrix $\boldsymbol{\Sigma}$ can be simply implemented by on-chip attenuators, e.g., single-port MZIs, to perform signal scaling. The unitary matrices $\boldsymbol{U}$ and $\boldsymbol{V}^*$ can be realized by a cascaded MZI triangular array [39]. The unitary group parametrization is given by,

$$
\boldsymbol{U}(N) = \boldsymbol{D} \prod_{i=N}^{2} \prod_{j=1}^{i-1} \boldsymbol{R}_{ij}(\phi_{ij}),
\tag{8}
$$

where $\boldsymbol{D}$ is a diagonal matrix with $\pm 1$ on its diagonal entries, and the 2-dimensional planar rotator $\boldsymbol{R}_{ij}(\phi_{ij})$ is an $n$-dimensional identity matrix where entries on $(i,i)$, $(i,j)$, $(j,i)$, $(j,i)$ are $\cos\phi_{ij}$, -$\sin\phi_{ij}$, $\sin\phi_{ij}$, $\cos\phi_{ij}$, respectively. Each rotator $\boldsymbol{R}_{ij}$ can be implemented by a 2×2 MZI that produces unitary interference of input light signals with a rotation angle $\phi$ as we show before.

## A.3  Optical Circuit Non-ideality

**Rotation Quantization.** Given the control resolution limits, we can only achieve discretized MZI rotation phase configurations. We assume the phases $\phi$ is uniformly quantized into $b$-bit within $[0, 2\pi]$,

$$\mathcal{Q}(\phi) = \texttt{Round}\Big(\frac{\phi \bmod 2\pi}{2\pi/(2^b - 1)}\Big)\frac{2\pi}{2^b - 1}. \tag{9}$$

We assume 8-bit quantization for phases of $\boldsymbol{U}$ and $\boldsymbol{V}^*$. For $\boldsymbol{\Sigma}$ matrices, we assume larger bitwidths can be affordable and practical.

**Phase shifter Variation.** Due to manufacturing error and thermal noises, the phase shift $\phi$ caused by a phase shifter is proportional to the device-related parameter, $\phi \propto \gamma$. Assume the real coefficient drifts from the theoretical value $\gamma$ by $\Delta\gamma$, the real phase shift will become $\tilde{\phi} = \frac{\gamma + \Delta\gamma}{\gamma}\phi$. We assume $\Delta\gamma \sim \mathcal{N}(0, 0.002^2)$. We denote this multiplicative error for all phase shifters as a diagonal $\boldsymbol{\Gamma}$ matrix, such that the non-ideal phase shifts become $\boldsymbol{\Phi}^v = \boldsymbol{\Gamma}\boldsymbol{\Phi}$.

**MZI Crosstalk.** Due to signal crosstalk, adjacent MZIs will have mutual coupling effects, such that the part of the phase shift $\phi$ for the $i$-th MZI will partially contribute to its neighboring MZI $\phi_j$ with a factor of $\omega_{i,j}$. This crosstalk effect can be simply modeled as coupling matrix $\boldsymbol{\Omega}$,

$$\begin{pmatrix} \phi_0^c \\ \phi_1^c \\ \vdots \\ \phi_{N-1}^c \end{pmatrix} = \begin{pmatrix} \omega_{0,0} & \omega_{0,1} & \cdots & \omega_{0,N-1} \\ \omega_{1,0} & \omega_{1,1} & \cdots & \omega_{1,N-1} \\ \vdots & \vdots & \ddots & \vdots \\ \omega_{N-1,0} & \omega_{N-1,1} & \cdots & \omega_{N-1,N-1} \end{pmatrix} \begin{pmatrix} \phi_0^v \\ \phi_1^v \\ \vdots \\ \phi_{N-1}^v \end{pmatrix}$$
$$\text{s.t. } \omega_{i,j} = 1, \quad \forall\, i = j$$
$$\omega_{i,j} = 0, \quad \forall\, i \neq j \text{ and } \phi_j \in \mathcal{P} \tag{10}$$
$$0 \leq \omega_{i,j} < 1, \quad \forall\, i \neq j \text{ and } \phi_j \in \mathcal{A}.$$

The diagonal factor $\omega_{i,j}, i = j$ is the self-coupling coefficient. $\omega_{i,j}, i \neq j$ is the mutual coupling coefficient [31, 20, 17]. We assume the self-coupling coefficient to be 1, and the mutual coupling coefficient is 0.005 for adjacent MZIs.

## B  Intractable Gradients for MZI Rotations

To optimize the MZI meshes, a straightforward idea is to use first-order methods to optimize all rotations phases $\boldsymbol{\Phi}^U$, $\boldsymbol{\Phi}^V$, and $\boldsymbol{\Phi}^\Sigma$. The analytical gradients for phases in unitary matrices are shown as,

$$\frac{\partial \mathcal{L}}{\partial \boldsymbol{R}_{ij}} = \big(\boldsymbol{D}\boldsymbol{R}_{n1}\boldsymbol{R}_{n2}\boldsymbol{R}_{n3}\big)^T \nabla_y \mathcal{L}\, x^T \big(\cdots \boldsymbol{R}_{32}\boldsymbol{R}_{21}\boldsymbol{\Sigma}\boldsymbol{V}^*\big)^T$$
$$\frac{\partial \mathcal{L}}{\partial \phi_{ij}} = \text{Tr}\bigg(\Big(\frac{\partial \mathcal{L}}{\partial \boldsymbol{R}_{ij}} \odot \frac{\partial \boldsymbol{R}_{ij}}{\partial \phi_{ij}}\Big)(e_i + e_j)(e_i + e_j)^T\bigg). \tag{11}$$

Therefore, it is prohibitively expensive to derive the analytical phase gradients, which is one of the key motivations for our subspace optimization method.

## C  Detailed Description of the Proposed Parallel Mapping Algorithm

We give a detailed description of our parallel mapping algorithm. Zeroth-order coordinate descent (ZCD) is used as an example. In line 4, we first derive and implement the optimal theoretical singular values and initialize $\boldsymbol{\Phi}^U$ and $\boldsymbol{\Phi}^V$ using the decomposed values. In lines 8-13, we use ZCD to alternately optimize phases in $\boldsymbol{U}$ and $\boldsymbol{V}^*$ under all non-ideal effects till convergence. The step size is strictly bounded by the smallest phase control resolution. Exponential decay is used to quickly reduce the learning rate to avoid divergence. Note that cosine-annealing will not work since the ZO descent will rapidly converge given its greedy search nature. Then at the end, due to the suboptimality in ZCD, we will perform OSP to find the current optimal singular values that minimize the mapping error given the trained $\boldsymbol{U}^T$ and $\boldsymbol{V}^{*,T}$.

**Algorithm 1:** Parallel Mapping with ZCD and OSP

---

**Input** :Mapping loss $\mathcal{L}^M$, mapping target $\boldsymbol{W}$, total iterations $T$, inner ZCD iterations $S$, step size decay factor $\beta$, ZCD step size upper bound $\delta\phi_u = \frac{2\pi}{2^{\min(b_l,b)}-1}$, ZCD step size lower bound $\delta\phi_l = \frac{2\pi}{2^{\min(b_m,b)}-1}$

1  $\delta\phi = \delta\phi_u$;
2  **for** *Weight block* $\boldsymbol{W}_{pq} \sim \boldsymbol{W}$ **do**
3     Step 1: SVD and Parametrization via Eq. (1);
4     $\boldsymbol{U}_{pq}(\boldsymbol{\Phi}_{pq}^U), \boldsymbol{\Sigma}_{pq}(\boldsymbol{\Phi}_{pq}^S), \boldsymbol{V}_{pq}^*(\boldsymbol{\Phi}_{pq}^V) = \mathtt{UP}\big(\mathtt{SVD}(\boldsymbol{W}_{pq})\big)$;
5     Step 2: ZCD on $\boldsymbol{U}_{pq}, \boldsymbol{V}_{pq}^*$;
6     **for** $t \leftarrow 0 \cdots T-1$ **do**
7       **for** $s \leftarrow 0 \cdots S-1$ **do**
8         Randomly sample a phase $\phi \sim \{\boldsymbol{\Phi}_{pq}^U, \boldsymbol{\Phi}_{pq}^V\}$;
9         **if** $\mathcal{L}_{pq}^M(\phi^{tS+s} + \delta\phi) < \mathcal{L}_{pq}^M(\phi^{tS+s})$ **then**
10          $\phi^{tS+s+1} \leftarrow \phi^{tS+s} + \delta\phi$;
11        **else**
12          $\phi^{tS+s+1} \leftarrow \phi^{tS+s} - \delta\phi$;
13        $\delta\phi \leftarrow \max(\delta\phi/\beta, \delta\phi_l)$;
14    Step 3: Optimal Projection on $\boldsymbol{\Sigma}_{pq}$;
15    $\boldsymbol{\Sigma}_{pq} \leftarrow \mathtt{diag}(\tilde{\boldsymbol{I}}^* \boldsymbol{U}_{pq}^* \boldsymbol{W}_{pq} \boldsymbol{V}_{pq} \tilde{\boldsymbol{I}})$;

**Output** :Converged phases $\boldsymbol{\Phi}^M$

---

# D Prove of Unbiased Gradient Approximation with Feedback and Feature Sampling

**Claim 2.** *Considering the $l$-th layer with input $x \in \mathbb{R}^N$ and pre-activation $y \in \mathbb{R}^M$, we denote the blocking weight matrix as $\boldsymbol{W} = \{\boldsymbol{W}_{pq}\}_{p,q=1,1}^{P=\frac{M}{k}, Q=\frac{N}{k}}$ and nonlinear activation as $\sigma$. During backward, we randomly sample the feedback matrix $\boldsymbol{W}^T \in \mathbb{R}^{N \times M}$ with a structured sparse mask $\mathcal{P}_{\boldsymbol{W}} = c_W(\mathcal{S}_W \otimes \mathbf{1})$. A similar sampling matrix $\mathcal{P}_x$ is applied to input features. The estimated gradients are unbiased, i.e., $\mathbb{E}[(\frac{\partial \mathcal{L}}{\partial \boldsymbol{\Sigma}})_{\mathcal{S}}] = \frac{\partial \mathcal{L}}{\partial \boldsymbol{\Sigma}}$.*

*Proof.* Given $\mathbb{E}[\mathcal{P}] = \mathbf{1}$, we have

$$
\begin{aligned}
\mathbb{E}[(\boldsymbol{W}_l^T)_{\mathcal{S}_{\boldsymbol{W}_l}}] &= \mathbb{E}[\boldsymbol{W}_l^T \odot \mathcal{P}_{\boldsymbol{W}_l}] = \boldsymbol{W}_l^T \\
\mathbb{E}[(\boldsymbol{x}_l^T)_{\mathcal{S}_{\boldsymbol{x}_l}}] &= \mathbb{E}[\boldsymbol{x}_l^T \odot \mathcal{P}_{\boldsymbol{x}_l}] = \boldsymbol{x}_l^T.
\end{aligned}
\tag{12}
$$

Then we can derive

$$
\begin{aligned}
\mathbb{E}[(\frac{\partial \mathcal{L}}{\partial y_l})_{\mathcal{S}_{\boldsymbol{W}_l}}] &= \mathbb{E}\Big[\sigma_l' \prod_{i=l+1}^{L-1} ((\boldsymbol{W}_i^T)_{\mathcal{S}_{\boldsymbol{W}_l}} \odot \sigma_i')(\boldsymbol{W}_L^T)_{\mathcal{S}_{\boldsymbol{W}_l}} \frac{\partial \mathcal{L}}{\partial y_L}\Big] = \frac{\partial \mathcal{L}}{\partial y_l} \\
\mathbb{E}[(\frac{\partial \mathcal{L}}{\partial \boldsymbol{\Sigma}_l})_{\mathcal{S}}] &= \mathbb{E}\Big[\boldsymbol{U}^* (\frac{\partial \mathcal{L}}{\partial y_l})_{\mathcal{S}_{\boldsymbol{W}_l}} (x_l^T)_{\mathcal{S}_{\boldsymbol{x}_l}} \boldsymbol{V}\Big] = \frac{\partial \mathcal{L}}{\partial \boldsymbol{\Sigma}_l}.
\end{aligned}
\tag{13}
$$

$\square$

# E Training Details

We implement ONN simulation, all models, and training logic in PyTorch 1.8.1. All experiments are conducted on a machine with an Intel Core i7-9700 CPU and an NVIDIA Quadro RTX 6000 GPU. For identity calibration, we set the epoch to 400 with an initial learning rate of 0.1, a decay rate of 0.99, and a phase resolution of 8 bit. For parallel mapping, we set the epoch to 300 with an initial learning rate of 0.1, a decay rate of 0.99, and a phase resolution of 8 bit. For subspace learning, we adopt AdamW as the optimizer with a learning rate of 0.002 and a weight decay rate of 0.01 for subspace learning from scratch. Epochs are set to 100 for MNIST, FashionMNIST training, 200 for CIFAR-10/100, and TinyImageNet. For subspace learning after mapping, we reduce the epoch to 20 and the learning

rate to 0.0002. We use cosine-annealing as the learning rate scheduler. When compared with prior on-chip learning protocols, we adopt the recommended settings for FLOPS and MixedTrn in [21, 17]. For FLOPS, the total epochs are set to 50, the initial learning rate is 2, and the gradient samples are set to 5. For MixedTrn, we train for 20 epochs, the mixed-training sparsity is set to 0.4, the parameter sparsity is set to 0.1, and the initial learning rate is set to 0.02. When compared with prior sampling methods, we apply uniform spatial sampling with expectation-maintained normalization for RAD [36]. For SWAT-U [38], we apply uniform spatial feature sampling without normalization and uniform weight matrix sampling with expectation-maintained normalization. Since we only perform efficient training, we turn off any sampling in inference.

## F  MZI Array Scaling

A single MZI array has a limited size due to its high area cost, e.g., up to 32 or 64. However, this is not an issue for our framework. Multi-core systems with small subarrays are trends for analog computing, which is the design concept of our accelerator in Figure 3. Multiple PTCs are interconnected to support a large tensor computation in parallel. Therefore, our system's performance will not be limited by the scale of a single PTC. Actually, partitioning a large tensor operation into small chunks is widely adopted and recently considered as a better solution than large array sizes due to noise robustness consideration.

We adopt 9×9 blocks based on the following considerations.

**Hardware practicality.** The largest commercial demonstration of optical neural chips is 32×32 so far. 9×9 is a practical, robust, and efficient setting according to recent experimental demonstrations.

**Robustness.** Larger MZI arrays will cause severe phase error accumulation effects. Cascaded phase error will cause non-trivial fidelity and robustness issues as block size increases. 9×9 is generally a robust design configuration when cascaded noises are still tolerable. Here we show a table of noise-induced errors (relative matrix distance) with various block sizes on a 256×256 weight matrix. Std. is calculated based on 20 runs. Phase shifter gamma noise std=0.002, crosstalk factor=0.005, quantization bitwdith=8-bit. We observe large array sizes are noise-sensitive in general.

| Blk size | 8 | 9 | 12 | 16 | 24 | 32 |
|---|---|---|---|---|---|---|
| Rel. Err. | 0.025 | 0.032 | 0.043 | 0.061 | 0.094 | 0.126 |
| std. | 2e-4 | 3e-4 | 3e-4 | 5e-4 | 9e-4 | 1e-3 |

Table 3: Relative matrix error with different MZI array sizes.

**ZOO Convergence.** IC and PM are zeroth-order optimization techniques. Each block indicates an optimization instance. A larger block size will have negative impacts on the optimization convergence and solution optimality, which is the intrinsic limitation of most zeroth-order optimizers. In the IC procedure, for relatively large block sizes, our ZO optimizers, unfortunately, will have solution quality degradation due to the curse of dimensionality and efficiency degradation due to low parallelism. Here we show how solution quality in identity calibration changes with various block sizes. 9×9 block is a good selection with high solution quality.

| Blk size | 8 | 9 | 12 | 16 | 24 | 32 |
|---|---|---|---|---|---|---|
| $(MSE^U + MSE^V)$/2 | 0.0135 | 0.013 | 0.03 | 0.039 | 0.04 | 0.045 |

Table 4: IC optimality with different array sizes.

**Parameter Space.** Subspace learning only optimizes the singular values while $U$ and $V$ are fixed. For an $N \times N$ weight matrix with $k \times k$ blocks, only $N^2/k$ singular values are trainable. Increasing the block size $k$ will decrease the parameter space. According to the experience from the field of structured/subspace neural networks, e.g., block-circulant neural nets, the block size is typically set to a number around 8. Here we add new results on L²ight-SL ($\alpha_W=\alpha_C=0.6$, $\alpha_D=0.5$) CIFAR-10 VGG8 with various block sizes. According to our experiments below, 16×16 blocks already show inadequate trainability due to overly small parameter space, leading to a clear accuracy drop. In conclusion, we recommend using multiple interconnected 9×9 PTCs for parallel computing, since this choice of 9×9 block balances both systematic performance, hardware complexity, robustness, and on-chip trainability.

| Blk size | 8 | 9 | 12 | 16 | 24 | 32 |
|---|---|---|---|---|---|---|
| Accuracy | 84.26 | 84.45 | 83.36 | 81.27 | 80.68 | 78.40 |

Table 5: Subspace learning accuracy with different block sizes.

# G Hardware Cost Evaluation

## G.1 PTC Energy Estimation

For simplicity, we count the number of PTC calls as the indicator to the total energy estimation of the PTC cluster. For example, we focus on a 2-D convolutional layer with kernel shape of $C_{out} \times C_{in} \times K \times K$, input feature size $B \times C_{in} \times H \times W$ output feature size of $B \times C_{out} \times H' \times W'$. We partition the unfolded weight matrix into $P \times Q$ blocks with size of $k \times k$ and assign each to a PTC. We have $P = \lceil \frac{C_{out}}{k} \rceil$ and $Q = \lceil \frac{C_{in} \times K^2}{k} \rceil$. Each PTC can utilize $k$ wavelengths to achieve parallel processing. Now we give detailed computation of energy breakdown per optimization iteration.

$$\textbf{Forward Energy} = C_{out} C_{in} K^2 B H' W'$$
$$\textbf{Backward Weight Energy} = 2\text{Tr}(\mathcal{S}_C^T \mathcal{S}_C) BPQ \tag{14}$$
$$\textbf{Backward Input Energy} = \text{Tr}(\mathcal{S}_W^T \mathcal{S}_W) BHW.$$

Note that in backward weight energy, we double the PTC call since the *in-situ* subspace gradient acquisition requires 2 PTC calls.

## G.2 Total Time Step Estimation

We assign $k$ electrical adders for each PTC to implement sequential cross-PTC reduction and parallel local accumulation. Each PTC call counts as one step, each partial product/gradient accumulation stage counts as one step, and the Hadamard multiplication in gradient computation also counts as one step. Given this assumption, we derive the time step as,

$$\textbf{Forward Step} = (Q - 1)_+ BH'W' + \lceil \frac{BH'W'}{k} \rceil$$

$$\textbf{Backward Weight Step} = 4\text{Tr}(\mathcal{S}_C^T \mathcal{S}_C) B$$

$$\textbf{Backward Input Step} = \begin{cases} \lceil \frac{C_{in}}{P} \rceil \lceil \log_2 2k \rceil \lceil \frac{1}{2} \max_q \left( \left( \sum \mathcal{S}_W(q,:) - 1 \right)_+ \right) \rceil BHW, & K > 1, \text{ stride} < K \\ \max_q \left( \left( \sum \mathcal{S}_W(q,:) - 1 \right)_+ \right) BH'W', & K = 1 \end{cases} \tag{15}$$

## G.3 WDM Dispersion Discussion

Theoretically, coherent photonic circuits will have slightly different phase responses to different working wavelengths. However, we claim that this frequency-specific phase shift has minimum impacts on our learning procedure.

**Negligible Dispersion.** Our PTC core is intentionally designed to have a small-scale, i.e., 9×9. Hence we require 9 wavelengths in our framework. This avoids too many wavelengths being used. Therefore, the spectrum range will be relatively small. Conservatively we assume 8 nm between the furthest two wavelengths. Based on the phase response equation, $\Delta\phi(\lambda) = 2\pi n_{eff}(\lambda) L / \lambda$, this leads to a maximum 1-2% phase difference for the furthest two wavelengths. On a small MZI array, this phase difference will only cause negligible transfer function drift. We simulate this effect when the weight block size is set to 9×9 and inject 1-2% dispersion-induced MZI phase response drift; the transfer matrix has 0.5% relative error and 0.5% mean square error. Compared with the gradient approximation error caused by our three-level sparse sampling, phase variation, and thermal crosstalk, shown in Fig. 8, this slight drift caused by WDM dispersion is negligible.

**High Non-ideality Tolerance.** Our experiments show that first-order subspace learning is very robust to all these gradient approximation errors. With all the above non-ideality, the approximated gradient directions are still well-aligned with the true gradients. The on-chip learning procedure works as expected even when WDM dispersion effects are considered. This effect can be considered in-situ when using WDM on MZI array training, therefore, the model can tolerate this non-ideal effect without inference accuracy degradation.

**Dispersion-free Devices.** In the literature, there are WDM dispersion-free MZI devices being proposed [12]. Within the 45nm range, the coefficient of phase shifters can be maintained. Thus, the phase response to 9 different wavelengths can be compensated to almost the same response. This

further shows that WDM dispersion is not a major concern for our assumed ONN architecture and proposed training flow.