# OpenReview forum: "L2ight: Enabling On-Chip Learning for Optical Neural Networks via Efficient in-situ Subspace Optimization"
_NeurIPS.cc/2021/Conference — NeurIPS 2021 Poster_

### Official Review · Reviewer_f3Zb · 2021-07-12

**Rating:** 7
**Confidence:** 3

**Summary:**

Optical Neural Networks (ONN) with their attojoule per Multiply Accumulate energy efficiency and sub-nanosecond latency are becoming useful implementation hardware for large scale deep learning models and datasets. Despite their efficiency and speed, ONNs suffer significant loss in accuracy due to manufacturing defects/non-ideal device controls and circuit noises. Simulating these non-idealities during software training is not scalable.

Training ONN on-chip is a hard problem that is exacerbated due to above factors. This paper tackles the problem by proposing a three-stage learning framework L2ight. The three stages are variation-agnostic calibration, alternate-projection based parallel mapping and multi-level sparse subspace learning.

L2ight enables scaling up on-chip ONN training, for the first time to models with 10M params (1000x scalable than baseline). This scalability is built on top of an efficient training algorithm utilizing multi-level sparsity and a restricted subspace optimization step which enables sufficient adaptability for on-device learning and task transfer. L2ight also achieves 30x higher efficiency than prior art. Lastly, the subspace learning step is agnostic to device/process variation and mapping noise enabling robust implementation of deep neural networks (DNN).


**Limitations And Societal Impact:**

The paper does not include any actual hardware measurements, it would be helpful if the authors specify it at some point as a limitation.

**Main Review:**

The paper presents a series of novel strategies at different stages of implementing a pre-trained DNN onto an ONN hardware followed by training for same/different task as pretrained. Optimal Singular Value Projection and batch break down of the mapping problem helps achieve fast and high accuracy mapping of pretrained/from scratch weights on to the ONN.  In-situ subspace gradient acquisition exploits reciprocity in optics for minimizing the cost of expensive gradient calculation. Ablation studies demonstrate the robustness of subspace learning step against device/mapping stage errors. Further, unique sampling strategies such as balanced topk and column sampling further help lower the number of hardware operations without cascading gradient variance across layers. Lastly, the paper also utilizes mini batch dropping (SMD) to improve training efficiency.

Section 4.1, paper describes the usage of number of PTC calls as the normalized energy indicator. Similarly, longest accumulation path is assumed to be the normalized latency/runtime indicator. Authors present a detailed breakdown of the energy/latency calculation in appendix based on the parameters of a given convolution layer. While these metrics are surely key factors impacting the energy consumption and latency of a DNN running on the ONN. It’s not clear that when accounting for other electrical components highlighted in Figure 3 (e.g., EO/GLB/Interconnects) if these factors will dominate the ONN energy and latency. Specifically, as highlighted in Section 3.4.1, part of gradient calculation (element-wise multiplication) is outsourced to electrical control units, is it obvious that the energy/latency cost of this outsourcing is negligible? It would be appreciated if the authors can justify this.

Table 2 shows a comparison of normalized PTC Energy and the normalized number of Steps, the ratios for both metrics are tightly aligned with a couple of exceptions, this goes back to my argument that while the proposed techniques help reduce the number of steps (PTC passes) it’s unclear if a hardware ONN system will receive energy efficiency benefits of the same order (30x). Similar observation holds true for Figure 13 with identical energy/step value variation. Consider consolidating this reduction of steps and normalized energy into a single parameter, e.g., no. of PTC operations, for which considerable reduction is demonstrated.

Section 4.2 and Figure 10 discuss a comparison of L2ight with prior work and demonstrates its superiority. A few clarifications about this data will be helpful, are the reported accuracy numbers the best achievable accuracy without any restrictions on number of training steps? Stats like the number of training steps/wall clock time for the different data points on Figure 10 would be helpful.

 Additionally, I would appreciate if the authors could clarify the following question. Where does the 1000x improvement in scalability come from? Is it simply from the batching of the optimization problem or is it due to efficient gradient calculation as described in Section 3.4.1? Also is there something unique about the L2ight flow that allows the demonstrated batching, or the batching could be applied to MixedTrn prior work to get similar scalability improvements?

The paper is well organized, and the detailed figures provide useful insights.

Typos and minor errors:
1.	Line 36, on-device training “has become/is becoming” an appealing trend towards …
2.	Line 60, 30x more efficient than prior “art”.
3.	Line 66, pi notation should have i=2 as starting and n as the max value. The same inversion is used in later equations representing U(n) in the paper and appendix.
4.	Figure 2: Stage 3 should have an accuracy check before finishing, it might be performed in block 3 (Update Σ), consider listing it out separately.
5.	Figure 9: Column sampling figure, the arrows start from x12 instead of x22 as in spatial sampling.
6.	Figure 14b: Legend appears wrong as it lists dataset name instead of DNN name.


**Time Spent Reviewing:**

5

---

> ### Author Response · Authors · 2021-08-06
> **Our sparse sampling techniques are hardware-aware with structured sparisity. Our efficiency improvement holds regardless of the proportion of electrical cost. Our scalability advantages mainly come from first-order optimization in the subspace learning stage.Thank you so much for going through the paper carefully and providing valuable feedback to our work**
>
> Thank you so much for going through the paper carefully and providing valuable feedback on our work! Please see our responses below:
>
> **C.1 Whether electrical components dominates latency/energy?**
>
> Thanks for your great question. Based on the system performance of the state-of-the-art photonic neural chip demonstration, the electrical memory and ADC still take a non-trivial proportion of the total energy cost and latency. OE/EO conversion and photonic chip execution are ultra-fast and energy-efficient. A good reference for systematic energy/latency breakdown can be found in recent multi-core optical accelerator designs [2][3][6]. Even though we are still not close to fully-optical accelerators, current electro-optic accelerators already demonstrate at least 2-order-of-magnitude higher efficiency and performance than commercial GPUs [1-6]. Their simulation/experimental data provide fair justification for ONN on-chip training applications.
>
> Besides the efficiency advantage of ONN compared with electrical computers, the most important point for our efficiency improvement techniques is that our three-level sampling methods directly _skip the pruned block_, which means the corresponding cost of _memory transaction, computation, control, and communication_ are together removed. Therefore, the sparsity we show in the paper can be directly translated to energy/latency improvement ratio regardless of whether the electrical part dominates the total cost. This _highly-structured sparsity_ guarantees that the 30$\times$ reduction we claim in the paper will nearly not be diluted even when the performance is bottlenecked by electrical peripherals.
>
> [1] Xingyuan Xu, Mengxi Tan, et al., “11 TOPS photonic convolutional accelerator for optical neural networks”, Nature 2021.
>
> [2] Farzaneh Zokaee, et al., “LightBulb: A Photonic-Nonvolatile-Memory-based Accelerator for
> Binarized Convolutional Neural Networks”, DATE, 2020.
>
> [3] Weichen Liu, et al., “HolyLight: A Nanophotonic Accelerator for Deep Learning in Data Centers”, DATE 2020.
>
> [4] Carl Ramey, “Silicon Photonics for Artificial Intelligence Acceleration” HotChips, 2020.
>
> [5] Bhavin J.Shastri, et al., “Photonics for artificial intelligence and neuromorphic computing”, Nature Photonics, 2021.
>
> [6] Mario Miscuglio, et al., “Photonic tensor cores for machine learning”, Applied Physics Review 2020.
>
> **C.2 Whether element-wise multiplication in the electrical controller is negligible?**
>
> Thanks for your great question. We would like to clarify this from three aspects.
> - In the singular value gradient computation for one 9x9 block, there will be only 9 scalar multiplication offloaded to the electrical controller for each input vector, which is fairly cheap compared with other tensor operations.
> - Second, multiple gradient vectors on a batch of inputs need to be accumulated in order to get the final gradient w.r.t. singular values. The element-wise multiplication and accumulation will actually be fused as a group of MAC operations. Hence, in practice, the element-wise multiplication will not introduce any extra burden to the electrical domain since gradient accumulation will utilize the MAC units anyway, i.e., the total cycle number will remain the same with element-wise multiplication.
> - Further, we would like to mention that the aforementioned fused multiply-add steps for element-wise multiplication and gradient accumulation are considered in our total step and energy calculation shown in the paper. According to our hardware cost profiling, the cost of this part is indeed negligible compared with other major costs.
>
> **C.3 How much total energy/latency can be saved by sparsifying PTC passes?**
>
> Thanks for your great question. Even though we refer to this operation as _PTC call_, it does not only include the execution of the photonic part. It refers to all the operations related to the PTC call, i.e., memory read/write, ADC/DAC, OE/EO, photonic controller, partial result broadcast and accumulation, etc. Any pruned block can eliminate the entire chunk of hardware cost no matter how much energy/latency proportion the electrical parts take. In other words, the claimed energy/latency reduction is a systematic estimation, not an estimation in the pure optical domain.
>
> Our sampling techniques are intentionally designed to have _structured sparsity_, which is addressed as our main advantage compared with previous hardware-unaware unstructured methods in Fig. 7 and Fig. 9.
> As a reasonable estimation, we assume each PTC call approximately costs similar energy and delay. Hence, our sparse sampling ratio will directly translate to overall system energy/latency improvement with a similar ratio, which is ~30x. This improvement will nearly not be diluted when the cost of electrical circuitry is counted. We will clarify this point in our manuscript to avoid confusion. Thanks a lot.
>
> **C.4 Consider merging PTC energy and steps as a single metric**
>
> Thanks for your great suggestions. Indeed the energy and step improvement are quantitatively quite similar. To simplify the figures and tables, we will merge them as a single metric, e.g., energy-delay product or just PTC operation as you suggested.
>
> **C.5 Clarify the data in Figure 10**
>
> Thanks for your suggestions. All training is performed with a fixed number of epochs. We did not select the highest accuracy during training. All energy/latency is evaluated by our profiler/counter. For our L2ight, the detailed training settings are shown in Appendix E. For FLOPS and MixedTrain, we use 50 epochs and 20 epochs, respectively. Longer epochs for FLOPS and MixedTrain on larger datasets will cost too much runtime and cause divergence. We will include more training settings details in the appendix. Thanks.
>
> **C.6 Where does the 1000x come from? Is it unique to L2ight?**
>
> Thanks for your great question. In our paper, we highlight the superiority of L2ight in Section 4.2, Line 262-276.
> The most important reason behind our scalability improvement is the subspace learning stage. We will give detailed reasons as follows.
> - On the hardware side:
>   - We do not require per-device monitoring and in-situ field detection, which makes our framework scalable.
>   - We directly utilize the MZI array in a reversed way to compute the gradients without costly auxiliary computing engines.
> - On the algorithm side:
>   - Our subspace learning leverages the on-chip reciprocity to obtain gradients in situ, which enables first-order gradient-based learning for modern DNNs with a relatively large number of parameters. This is the fundamental advantage of our method. Compared with the prior zeroth-order method, this feature avoids the curse of dimensionality and leads to this 1000x improvement.
>   - Note that our scalability benefits do not come from pre-trained models or efficient parallel mapping but mainly from the first-order optimization algorithm (subspace learning) itself. The first and second stages (IC and PM) are 3-order-of-magnitude cheaper than the third learning stage, thus the batched zeroth-order optimization is not the main reason for our scalability superiority.
>   - Indeed, parallel mapping can take good advantage of pre-trained models which can help fine-tuning procedures converge faster and better (But this is not a must for on-chip training). Besides, our three-level sparse sampling techniques can reduce the cost in the subspace learning stage and further boost efficiency.
>
> **C.7 Minor issues**
>
> Thanks for your careful reading and kind suggestions.
> - All typos in the text, figures, and tables will be corrected in our manuscript.
> - Line 66, The equation follows the convention to construct MZI arrays widely used in previous papers [Shen+, NaturePhotonics’17] [Zhao+, ASP-DAC’19]. We double-checked and confirmed that the start and end indices are correct. We will give a detailed discussion in our supplementary.
> - We will fix the flow chart and add the accuracy check as suggested.
>
> **C.8 Limitation discussion**
>
> Thanks for your suggestions.
> Our work demonstrates the possibility to achieve an orders-of-magnitude boost in the ONN in-situ training scalability and efficiency. We fully agree that the results shown in the current manuscript have the following limitations.
> - Our current training results and energy/latency estimation are based on software simulation without real chip demonstration. Actually, in order to emulate the non-ideality on real chips, we have hardware non-ideality modeling during simulation, including phase variation, manufacturing phase bias, device crosstalk, limited device control resolution, etc. In the near future, we plan to validate the effectiveness and robustness of our learning framework on our optical neural chips as a prototype demonstration.
> - Besides the optical matrix multiplication engine, currently, other peripheral components, e.g., memory, interconnects, and nonlinearity, are assumed to be in the electrical domain. The systematic performance is still limited by those electrical parts. In the future, we envision a fully-optical solution that can integrate advanced photonic nonvolatile memory, optical interconnects, and fully-optical nonlinearity for higher performance.
>
> We will add a formal limitation discussion section in our manuscript to clarify those points. Thanks.
>
> We wish that our response has addressed your concerns, and turns your assessment to the positive side. If you have any questions, please feel free to let us know during the rebuttal window. Thank you very much! We appreciate your suggestions and comments!

---

> > ### Comment · Reviewer_f3Zb · 2021-08-23
> > **Rebuttal response**
> >
> > I appreciate the detailed response from the authors to my concerns. Overall, I am satisfied with the responses from the authors. I do have one minor concern and I would appreciate if the authors could clarify on that.
> >
> > Like the response to C6, the major savings are explained by in-situ gradient calculation (hardware and algorithm) as well as the efficient sparse sampling techniques which reduce the number of operations across the system (memory reads, ADC/DAC, OE/EO, PTC calls and accumulation).
> >
> > While it makes sense that the authors have correctly modelled the savings from the sampling techniques, could the authors clarify as to how the solutions that do not utilize the proposed gradient calculation algorithm get their gradients? Also how is the cost modelled for the state-of-the-art gradient calculation?
> >
> > Another question on the same topic, Figure 11, shows a breakdown of energy savings from the different components employed by L2ight. It is evident that the 30x energy savings can be broken down as 3x for the sampling techniques and 10x for the in-situ gradient calculation. It is important that the authors have modelled this 10x improvement keeping in mind the significant cost of electrical components as agreed upon in the response to C1. While the in-situ techniques avoid expensive gradient calculation, they do increase the computational load on the ONN. If the authors could give a detailed description of this 10x improvement cost, I do not have any other concerns about accepting this paper.

---

> > > ### Author Response · Authors · 2021-08-23
> > > **Prior methods are derivative-free method whose cost can be simply counted by number of forward PTC calls. Our 30x improvement comes from sparse learning and**
> > >
> > > Thank you so much for providing valuable feedback on our work! Please see our responses below:
> > >
> > > # **How did previous methods obtain gradients? And how did the author model their cost?**
> > >
> > > * As we mentioned in the related work section (Line 44), previous ONN training protocols include brute-force tuning (BFT), evolutionary algorithm (particle-swarm optimization PSO), FLOPS, MixedTrain. All those are zeroth-order (derivative-free) algorithms.
> > >   * BFT uses greedy search on each phase in a sequential manner. It does not have any gradient estimation. It is very slow in general. So we do not compare with it.
> > >   * PSO searches for a better solution by perturbing and evolving the populations without gradient estimation. It is not stable and quite slow. We do not compare with it.
> > >   * In Table 1, we also review a prior work named the adjoint variable method (AVM). which leverages the property of light field and solves the inverse design problem to obtain the analytical gradients to each control variable. This method requires per-device light field monitoring with special hardware supports, which is not efficient or scalable in practice. Therefore, we do not compare with this method in our discussion.
> > >   * One of the SOTA methods is **FLOPS**, which uses the zeroth-order gradient (based on the numerical difference) as an approximation to the first-order gradient. Specifically, it perturbs all phases along a sampled direction and evaluates the loss function. Based on the difference in the loss function, it can approximate the directional gradient (zeroth-order gradient), and apply gradient descent based on this gradient.
> > >   * Another SOTA method is **MixedTrain**. It applies coordinate descent on a sparse subset of phases. Specifically, it fixes most of the phases and perturbs each non-fixed phase one by one to find a lower loss value.
> > >
> > > As we clarified above, except AVM, those prior work mainly depends on **multiple forward propagations** to estimate the device tuning direction without any "back-prop" or "first-order gradient calculation" process. Therefore, the cost model for those methods is quite simple. **We just need to count the number of forward calls of PTCs** as the cost estimation. The PTC forward cost model for FLOPS and MixedTrain is the same one used in our L2ight framework, shown in **Appendix F.1 _Forward Energy_ and F.2 _Forward Step_**.
> > >
> > > # **How to understand the 30x improvement?**
> > > Thanks for your great question on this important contribution.
> > > The first point we want to address here is the _comparison objects_.
> > >   * Compared with first-order sparse training methods in the machine learning community, e.g., RAD, SWAT-U. Our _L2ight_ has a 30x improvement in energy/step cost.
> > >   * Compared with previous SOTA ONN on-chip training protocols, e.g., FLOPS, MixedTrain, we have 1000x improvement in scalability.
> > >
> > > When we discuss the energy/step reduction in Figure 11, we have already narrowed our scope in sparse training method comparison with RAD and SWAT. We are not comparing with ONN training protocols (FLOPS, MixedTrain) in this context since ResNet and VGG are not even tractable for them (recall our 1000x scalability advantages shown in Figure 10).
> > >
> > > Once this point is clear, now we can move to the explanation of the 30x energy/step improvement.
> > >
> > > In Figure 11, we have two bars for our method. (1) _L2ight-SL_ is training from scratch using our subspace learning method and three-level sparse sampling. (2) _L2ight_ is the full three-stage flow from _identity calibration (IC)_ -> _parallel mapping (PM)_ -> _subspace learning (SL)_.
> > >   * By applying our subspace learning with three-level sparse sampling, our L2ight-SL can improve efficiency by 3-4X compared with baseline, RAD, and SWAT-U. This shows the effectiveness and efficiency of our efficient sparse learning algorithm introduced in Section 4.3.1. The reviewer has a very correct understanding of this 3x improvement.
> > >   * However, the reviewer seems to misunderstand the source of the extra 10x improvement. We would like to clarify that it is not due to in-situ gradient computation. In-situ gradient computation enables first-order ONN training, leading to 1000x scalability improvement compared with prior zeroth-order training protocols (FLOPS, MixedTrain). **The extra 10x improvement of our L2ight comes from parallel mapping (PM)**. As we summarized in Section 4.2 Lines 266-272, PM can fully leverage the pre-trained weights and make the subsequent on-chip subspace learning procedure **converge faster and better**, proved by the data in Figure 13. With parallel mapping, ONNs can gain ~10x faster convergence with higher accuracy than training from scratch (L2ight-SL). Therefore, the extra 10x energy/step saving is the result of faster and better convergence by using parallel mapping, not from in-situ gradient computation.
> > >
> > > We would like to address the logic flow here:
> > > * Our in-situ gradient calculation is the foundation of our superior scalability compare with FLOPS and MixedTrain, shown in Line 256-261. Without this in-situ algorithm, none of our experiments on VGG and ResNet is possible.
> > > * We are not satisfied by this 1000x scalability, we want to further improve the efficiency of the subspace learning stage (Line 164-166). Therefore we propose three-level sparse sampling in Section 4.3.1. Our L2ight-SL with sparse sampling achieves 3-4x improvement compared with other efficient training methods (RAD, SWAT-U).
> > > * Then we combine our parallel mapping with subspace learning to accelerate the convergence. Our complete flow _L2ight_ shows the best results and fastest convergence, leading to the ultimate 30x energy/step reduction.
> > >
> > >
> > > We will add those explanations in the revision to clarify these two points and remove any potential misunderstandings by readers.
> > > We wish that our response has addressed your concerns, and turns your assessment to the positive side. If you have any questions, please feel free to let us know during the rebuttal window. Thank you very much! We appreciate your suggestions and comments!

---

> > > > ### Comment · Reviewer_f3Zb · 2021-08-26
> > > > **Baseline gradient modeling and breakdown of 30x energy efficiency benefit explained correctly**
> > > >
> > > > Appreciate the prompt and detailed response from authors. Thanks for the detailed breakdown of gradient calculation approach or lack thereof for each of the baselines.
> > > >
> > > > The clarification of 30x energy efficiency in terms of contributing factors PM instead of in-situ gradient calculation is very helpful. I do not have any other concerns with this paper. Updating my rating accordingly.

---

> ### Author Response · Authors · 2021-08-22
> **A kind reminder of our responses to your comments. Thank you very much.**
>
> Thanks so much for your careful reading of our paper and your valuable questions and comments. This is a **kind reminder of our responses** to your review comments. If you have any further concerns or questions after reading our responses, please feel free to let us know during the rebuttal window. We wish that our response has addressed your concerns, and turns your assessment to the positive side. Thank you very much! We will appreciate any follow-up suggestions!

---

### Official Review · Reviewer_EGjz · 2021-07-15

**Rating:** 8
**Confidence:** 4

**Summary:**

The paper proposes a calibration and training method for optical neural networks, and more precisely suited for Mach Zhender interferometers arrays. The method is claimed to be scalable, and based on easily accessible optical quantities on chip.


**Ethical Concerns:**

None, as this is low level hardware training.

**Limitations And Societal Impact:**


What is not fully clear is the scaling. The paper claims up to 10M parameters, but remains restricted to 9x9 MZI arrays, if I understand well. This means that only relatively small unitaries are performed. I would really have liked to see a study of the scaling with respect to the size of the array, which is a major issue in future ONN. Another issue that could have been detailled is the WDM : how does the algorithms deals with non-idealities of the circuit with respect to multiple wavelength travelling through the same circuit and sensing a slightly different matrix.

Potential negative social impact: None, as this is low level hardware training.

**Main Review:**

The paper is overall well organized, and compared with the state of the art. The method is well described, and the performances are well backed through simulations of a 9x9 photonic tensor core. The results are evaluated in terms of efficiency, memory and number of step reduction.

The significance is potentially high, optical implementations of ONN using MZI remains highly dependent on the efficient tuning of the interferometer to a given transformation and/or to direct training, and subject to many sources of imperfection. All these aspects are taken into account and the algorithm appears to give a satisfactory answer to all these potential pitfalls at once. The method is also quite interesting, based on injecting the circuit from both sides iteratively to benefit from its time reversal symmetry.


**Time Spent Reviewing:**

1

---

> ### Author Response · Authors · 2021-08-06
> **Our multi-core photonic accelerator consists of multiple PTCs with a small MZI array size for parallel computing. We added discussions on the scaling of array sizes. WDM dispersion has minimum impacts on the effectiveness of our learning framework. Thank you so much for going through the paper carefully and providing positive and valuable feedback to our work**
>
> Thank you so much for going through the paper carefully and providing valuable feedback on our work! Please see our responses below:
>
> **C.1 Scaling of MZI array size**
>
> Thanks for your comments. For a _single_ MZI array, scalability indeed is a general concern. However, this is not an issue for our framework. _Multi-core systems with small subarrays are trends for analog computing_. Our accelerator in Fig. 3 is a multi-core electro-optic accelerator. Multiple 9$\times$9 PTCs are interconnected to support a large tensor computation in parallel. Therefore, our system’s performance will not be limited by the scale of a single PTC. Actually, partitioning a large tensor operation into small chunks is widely adopted and recently considered as a better solution than large array sizes. This trend is clear both in photonic and ReRAM accelerator designs (Even for compact ReRAM arrays, 8x8 now is preferred to previous designs of 128x128 or 32x32 arrays due to robustness consideration.).
>
> We adopt 9x9 blocks in our discussion based on the following considerations.
> - From the hardware side
>   - _Practicality:_ The largest commercial demonstration of optical neural chips is 32x32 so far. 9x9 is a practical, robust, and efficient setting according to recent experimental demonstrations from both academia and startup companies.
>   - _Robustness:_ Larger MZI arrays will cause severe phase error accumulation effects [Shen+, NaturePhotonics’17]. Cascaded phase error will cause non-trivial fidelity and robustness issues as block size increases. 9x9 is generally a robust design configuration when cascaded noises are still tolerable.
> Here we show a table of noise-induced errors (relative matrix distance) with various block sizes on a 256x256 weight matrix. Std. is calculated based on 20 runs. Gamma noise std=0.002, crosstalk factor=0.005, quantization bitwdith=8bit (same as used in the paper). Large array sizes are noise-sensitive in general.
> | block size | 8 | 9 | 12 | 16| 24 | 32 |
> |:----------------:|:------------:|:------:|:------:|:------:|:----:|:------:|
> |  relative errors (std.) | 0.0254(0.0002) |	0.0321(0.0003) | 0.0433(0.0003) | 0.0607(0.0005) | 0.0938(0.0009) | 0.1259(0.0011)|
>
>   - _Wavelength Utilization:_ We are using the wavelength-division multiplexing (WDM) technique to support parallel computing. To benefit from larger arrays, more wavelengths are required, which will cause non-trivial hardware overhead, especially the complexity of control and filtering circuitry. Typically, around 8-16 wavelengths are practical in analog optical computing [1].
>   - _Efficiency:_ Given that typical convolutional kernels are 3x3, a size-9 block does not need to pad the matrix with an extra block. Thus 9x9 is an efficient design configuration.
> - From the algorithm side
>   - _ZOO Convergence:_ Identity calibration and parallel mapping are zeroth-order optimization techniques, each $k\times k$ block indicates an optimization instance. A larger block size $k$ will have negative impacts on the optimization convergence and solution optimality, which is the intrinsic limitation of most zeroth-order optimizers. In the identity calibration procedure, for relatively large block sizes, our ZO optimizers, unfortunately, will have solution quality degradation due to the curse of dimensionality and efficiency degradation due to low parallelism.
> Here we show how solution quality in identity calibration changes with various block sizes. 9x9 block is a good selection with high solution quality.
> | block size | 8 | 9 |12 | 16| 24 | 32 |
> |:----------------:|:------------:|:------:|:------:|:------:|:----:|:------:|
> | (MSE$^U$+MSE$^V$)/2 | 0.0135 | 0.013 | 0.03 | 0.039 | 0.04 | 0.045 |
>   - _Parameter Space:_ Subspace learning only optimizes the singular values while U and V are fixed. For an $N\times N$ weight matrix with $k\times k$ blocks, there are only $N^2/k$ trainable singular values. Increasing the block size $k$ will decrease the parameter space. According to the experience from the field of structured/subspace neural networks, e.g., block-circulant neural nets, the block size is typically set to a number around 8.
> Here we add new results on L2ight-SL ($\alpha_W$=$\alpha_C$=0.6, $\alpha_D$=0.5) CIFAR-10 VGG8 with various block sizes. According to our experiments below, 16x16 blocks already show inadequate trainability due to overly small parameter space, leading to a clear accuracy drop. Our 9x9 block is a reasonable setting.
> | block size | 8 | 9 | 12 | 16| 24 | 32 |
> |:----------------:|:------------:|:------:|:------:|:------:|:----:|:------:|
> | Accuracy | 84.26 | 84.45 |83.36 | 81.27 | 80.68 | 78.40 |
>
> To conclude, given the hardware constraints and algorithmic effectiveness considerations, further increasing the PTC block size is neither efficient nor robust. In contrast, we recommend using multiple interconnected 9x9 PTCs for parallel computing, since this choice of 9x9 block balances both systematic performance, hardware complexity, robustness, and on-chip trainability.
>
> We will add the above discussions on the scaling in the revision. Thanks very much for your suggestions.
>
>
> **C.2 WDM dispersion concerns**
>
> Thanks for your great question. Theoretically, coherent photonic circuits will have slightly different phase responses to different working wavelengths. However, we claim that this frequency-specific phase shift has minimum impacts on our learning procedure.
> - Our PTC core is intentionally designed to have a small-scale, i.e., 9x9. Hence we require 9 wavelengths in our framework. This avoids too many wavelengths being used. Therefore, the spectrum range will be relatively small. Conservatively we assume 8 nm between the furthest two wavelengths. Based on the phase response equation, $\Delta\phi(\lambda)=2\pi n_{eff}(\lambda)L/\lambda$, this leads to a maximum 1-2% phase difference for the furthest two wavelengths. On a small MZI array, this phase difference will only cause negligible transfer function drift. We simulate this effect when the weight block size is set to 9x9 and inject 1-2% dispersion-induced MZI phase response drift; the transfer matrix has ~0.5% relative error and ~0.5% mean square error. Compared with the gradient approximation error caused by our three-level sparse sampling, phase variation, and thermal crosstalk, shown in Fig. 8, this slight drift caused by WDM dispersion is negligible.
> - Our experiments show that first-order subspace learning is very robust to all these gradient approximation errors. With all the above non-ideality, the approximated gradient directions are still well-aligned with the true gradients, as shown in Fig. 8. The on-chip learning procedure works as expected even when WDM dispersion effects are considered. This effect can be considered in-situ when using WDM on MZI array training, therefore, the model can tolerate this non-ideal effect without inference accuracy degradation.
> - In the literature, there are WDM dispersion-free MZI devices being proposed [2]. Within 45nm range, the $V_{\pi}$ coefficient of phase shifters can be maintained. Thus, the phase response to 9 different wavelengths can be compensated to almost the same response. This further shows that WDM dispersion is not a major concern for our assumed ONN architecture and proposed training flow.
>
> We will add the WDM discussion in the supplementary. Thanks very much for your suggestions.
>
> [1] Xingyuan Xu, Mengxi Tan, et al., “11 TOPS photonic convolutional accelerator for optical neural networks”, Nature 2021.
>
> [2] Nicolas Dupuis, Benjamin G. Lee, et al., “Design and Fabrication of Low-Insertion-Loss and Low Crosstalk Broadband 2x2 Mach-Zehnder Silicon Photonic Switches”, JLT.

---

> > ### Comment · Reviewer_EGjz · 2021-08-17
> > **Thank you for your answer.**
> >
> > Regarding my 1st comments, the author’s answer is satisfactory. Regarding the scaling, I get the point (although I am still hopeful to see larger scale unitary operations) and furthermore I appreciate that the error does not scale too badly with the size of the networks. What would be interesting to get also is the computing time scaling with the block size. I am also happy with their answer regarding dispersion.

---

### Official Review · Reviewer_jcqb · 2021-07-18

**Rating:** 7
**Confidence:** 4

**Summary:**

This paper introduces L2ight, a framework for mapping a pre-trained machine learning model to an optical neural network implemented with MZIs. The framework also supports on-chip learning: this is done by fine-tuning the implemented weights to compensate for manufacturing imperfections in the chip, or to perform transfer learning from one domain to another.
To do so, L2ight first performs an identity calibration step, then map the pre-trained weights, and finally uses subspace learning to enable efficient model fine-tuning to a specific task or to a specific chip. The paper provides extensive empirical evidence motivating this approach, focusing on compute vision applications. It is able to deal with architectures orders of magnitudes larger than the current state-of-the-art.

**Limitations And Societal Impact:**

As I mentioned above, the authors currently does not discuss at all the limitations of their work. There needs to be a section or a discussion in the conclusion doing so, in particular regarding: (1) transfer to real hardware; (2) use of pre-trained networks only, with no demonstrated ability to train from scratch. This could also be done by more clearly framing the motivations for this work initially.

**Main Review:**

*For ease of answering, I have annotated my various points with O.1/Q.1/etc...*

This is overall a solid paper, introducing a framework significantly improving the state-of-the-art, and with well thought experiments. The paper is very dense, but the authors have made a noticeable effort to make it clear.

One significant flaw however sticks out: the lack of discussion of the limitations of the framework proposed. This discussion is not optional, and is an integral part of a good scientific study. Currently, it is completely missing, even in a very short form. This leads to some part of the paper being borderline misleading: no actual training from scratch is ever performed in the experiments, and none of the experiments are on actual hardware.

**Because of the absence of a discussion around limitations of the method, this paper is marginally below the acceptance treshold (5)**. However, if such a discussion of limitations is added during the rebuttal, I would be willing to increase my review to an accept or a clear accept (7/8).

### Originality

This paper builds upon a body of work on optical neural network, and more specifically on the question of mapping a pre-trained network to the weights implemented on the chip by MZIs. It is well placed in the literature, and makes a clear, novel contribution.

### Quality

The simulated experiments provided are very thorough and all concrete claims are well supported.

However, limitations of the study are not outlined at all.
**Q.1**: One significant issue is that all experiments are limited to simulations, and the scheme is never implemented on a real optical chip. In itself, this is acceptable: this is cutting-edge hardware, which may be difficult to source and use. However, this is never explicitly mentioned in the paper. This is a significant limitation that needs to be discussed, in particular as inaccurate simulations could potentially worsen results (this is mentioned by the authors for other frameworks, l33).

**Q.2**: A small issue is that the various methods introduced in section 3 lacks any indication of scaling. It's not clear at which scale (size of matrices) each experiment is performed, and it's not clear how the compute/time cost scales with the size of the matrix. Explicitly introducing an O(n/log(n)/n^2) quantization of the cost of the different steps would be interesting, even if the scaling is only evaluated empirically. This could be added to the supplementary.

### Clarity

Despite being quite dense, the writing is always clear, and the authors have made a welcome effort of using bold/italics to highlight and structure information. The structure is easy to follow and provides a good narrative.

However, once again, the lack of discussion around the limitations of the proposed method hurt the paper.
**C.1**: Some of the sentences in the paper reads more like a marketing copy than a scientific paper. "unleashing the power of optics for real on-chip intelligence" (l12), "pushes this emerging field from intractable to scalable and further to efficient for next-generation self-learnable photonic neural chips" (l17) are two such significant claims made in the abstract, which are never backed in the paper. This is worsened by the absence of any discussion of the limitations of the approach, which is unacceptable for a conference paper.

**C.2**: I feel like the concept of "on-chip learning" needs to be clarified. Although a person having ordinary skills in the art of ONNs may understand what is meant, this is not the case for the more general machine learning oriented audience of NeurIPS. In particular, "on-chip learning" brings expectations that: a) learning (backward and optimization) operations are actually performed by the chip; b) neural networks are learned from scratch, with randomly initialized weights. Here, what the authors propose is more akin to fine-tuning: a pre-trained model is fine-tuned from its trained configuration to better fit the imperfections of the chip, and potentially to transfer to another domain (section 4.3.2). Currently, none of the experiments provided prove that learning can occur from scratch. This should be clearly indicated.

The following two points are small recommendations: they have not influenced my scoring of the paper.
**C.3**: The paper is very dense, and so are the figures. Because they are quite small, they can be really challenging to parse sometimes. I feel like they sometimes contain unnecessary information (such as the DaDiannao ASIC in Figure 1.a), which is never mentioned/discussed in the main text). However, the authors have clearly made an effort to make the figures more readable with annotations, so I do not hold this point strongly against them.

**C.4**: The paper could use a final proof-reading pass. There are a number of typos, such as  "*the* optical neural networks (ONNs)" (l25), "Recently, on-device training *becomes*" (l36), and repeated equations l117 and l119. Once again, this is fairly minor and I do not account for it in my current score, expecting that this will be fixed in the rebuttal phase.

### Significance

L2ight is a significant improvement over the state-of-the-art, building upon and improving previous works clearly.


**Time Spent Reviewing:**

5

---

> ### Author Response · Authors · 2021-08-06
> **We clarify that our framework supports on-chip training from scratch with good scalability. We added discussion on limitation, matrix size, and stage complexity. Thanks a lot for your valuable feedback.**
>
> Thank you so much for going through the paper carefully and providing valuable feedback on our work!
> Please see our responses below:
>
> **Q.1 Discussion on limitation.**
>
> Our work demonstrates the possibility to achieve an orders-of-magnitude boost in the ONN in-situ training scalability and efficiency. Our framework supports on-chip training from scratch, which is discussed in the response to C.2.
>
> Besides, we fully agree that the results shown in the current manuscript have the following limitations.
> - Our current training results and energy/latency estimation are based on _software simulation_ without real chip demonstration. Actually, in order to emulate the non-ideality on real chips, we have hardware non-ideality modeling during simulation, including phase variation, manufacturing phase bias, device crosstalk, limited device control resolution, etc. In the near future, we plan to validate the effectiveness and robustness of our learning framework on our optical neural chips as a prototype demonstration.
> - Besides the optical matrix multiplication engine, currently, other peripheral components, e.g., memory, interconnects, and nonlinearity, are assumed to be in the electrical domain. The systematic performance is still limited by those electrical parts. In the future, we envision a fully-optical solution that can integrate advanced photonic nonvolatile memory, optical interconnects, and fully-optical nonlinearity for higher performance.
>
> We will add a formal limitation discussion section in our manuscript to clarify those points. Thank you.
>
> **Q.2 Scaling of matrix size and complexity analysis of each stage**
>
> Thanks for your comments.
>
> In terms of matrix size, all matrix sizes follow the standard VGG-8, ResNet-18. All convolutional kernels are flattened into 2-D tensors using the im2col algorithm. We partition the weight matrix into a bunch of 9$\times$9 blocks for parallel processing. Our multi-core architecture, shown in Fig. 3, supports parallel block forward to implement one large neural layer.
>
> In terms of the complexity of our three-stage training flow, we will show the cost breakdown and complexity analysis as follows.
>
> _Assumption_: total optimization step in identity calibration, parallel mapping, and subspace learning is $T_1$, $T_2$, and $T_3$, respectively. There are $L$ layers, each layer including an $N\times N$ weight matrix partitioned into multiple $k\times k$ blocks.
> - The first stage is identity calibration. Each step performs coordinate descent on $k(k-1)$ phases for each $k\times k$ weight block. All $LN^2/k^2$ blocks are optimized in parallel, i.e., batched parallel zeroth-order optimization. The total step is $2k(k-1)T_1$. The total PTC call is $2LN^2(k-1)T_1/k$, approximately $2LN^2T_1$.
> - The second stage is parallel mapping. Each step performs coordinate descent on $k(k-1)$ phases for each $k\times k$ weight block. All $LN^2/k^2$ blocks are optimized in parallel, i.e., batched parallel zeroth-order regression. The optimal singular value projection (OSP) costs 3 steps. The total step is $2k(k-1)T_2+3$. The total number of PTC call is $2LN^2(k-1)T_2/k+3$, approximately $2LN^2T_2$.
> - The third stage is subspace learning. We assume the feature map size is $H\times W$ with batch size $B$. The detailed complexity analysis is given in _Appendix F_. The total step is approximately $T_3LNBHW/k$.
>
> According to our performance profiling breakdown, stages 1 and 2 in total have 3-order-of-magnitude cheaper costs than that of the subspace learning stage. Intuitively understanding, batched parallel regression is deterministic and data-independent, which is fundamentally faster and easier to solve than stochastic and data-driven subspace learning. That is why our efficiency optimization techniques mainly focus on the third stage as analyzed in Line 165-166. In the supplementary, we will add a formal complexity discussion for each stage with numerical examples for readers to better understand our flow as suggested. Thanks a lot.
>
>
> **C.1 Writing style of claims**
>
> Thanks for your great suggestions. We will rephrase our claims and conclusions, and make them more concrete, specific, and well-supported. We will clarify our limitations since we are currently based on simulation results and have not demonstrated our framework on real photonic neural chips. Here we would like to address that the 1000x scalability improvement comes from our subspace learning methodology. The 30x efficiency improvement comes from our three-level sparse sampling techniques. Previous on-chip training protocols can only handle toy examples without thorough consideration of on-chip noises, while our method extends the ONN on-chip learnability to modern neural networks with order-of-magnitude larger scales, and we can well-handle various device-level non-idealities and noises during training. The aforementioned numbers have experimental justification in Table 2 and Fig. 10, 13.
>
>
> **C.2 Clarify on-chip training. Does it support training from scratch?**
>
> Thanks for your valuable comments and great questions.
>
> _On-chip training_: As shown in Fig 2, if the flow starts from a pre-trained model, the pre-training is assumed to happen or already be done on offline-chip electrical engines, e.g., GPUs. (Typically the pre-trained weights for popular DNN models are readily available as public checkpoints without the need to retrain by ourselves). The identity calibration, parallel mapping, and subspace learning (forward, backward, gradient calculation, parameter update, etc) all happen on our multi-core photonic accelerators as shown in Fig. 3.
> SRAM, electrical control, photonic matrix multiplication, OE/EO conversion, interconnects are assumed to be all integrated on chip to support fully on-chip training.
>
> _Training from scratch:_
> Our three-stage flow indeed performs better if pre-trained weights are readily available, shown in Fig. 13. However, to clarify, a pre-trained model is _not a must_ for our framework to work, indicated by the <pretrain?> conditional branch in Fig. 2. Our framework supports training from scratch and pure on-chip learning. Our scalability benefits do not come from pre-trained models but mainly from the first-order optimization algorithm itself, as we analyzed in Section 4.2 Line 272.
>
> We have shown experimental results when doing ONN on-chip training from scratch.
> In Table 2, $L^2ight$-$SL$ series directly use subspace learning (SL) to do first-order on-chip training from scratch. $L^2ight (IC\rightarrow PM \rightarrow SL)$ adopts the full flow with pretraining $\rightarrow$ calibration $\rightarrow$ mapping $\rightarrow$ subspace training. The subspace learning (SL) procedure is the most important procedure in our framework since it enables in-situ gradient calculation and first-order optimization, which fundamentally makes on-chip training more scalable and efficient, compared with prior pure zeroth-order on-chip tuning.
>
> We would like to mention that parallel mapping can better leverage pre-trained models to make the SL stage converge faster and better as shown in Fig.13 and claimed in Line 266, but this PM stage is _not a must_. Also, on-chip transfer learning here in the paper is mainly demonstrated to justify that the restricted learning space is large enough to migrate to other datasets and tasks, which we believe is an interesting point to discuss but still _not a must_ in our flow. Note that without transferring, _training from scratch is fully supported by our framework_. With our sparse learning techniques, the training efficiency and inference accuracy outperform the prior method. The scalability superiority still holds when training from scratch, as shown in Table 2 and Figure 10, 11.
> We will clarify and emphasize those points in the paper. Thanks a lot.
>
> **C.3 Dense format**
>
> Thanks for your understanding and kind suggestions.
> We will try to reorganize our figures and tables and try to make our methods and results clearer and easier to read.
>
> **C.4 Typos**
>
> Thanks for your careful reading. We will fix all typos and polish our paper. Thanks.
>
> We wish that our response has addressed your concerns, and turns your assessment to the positive side. If you have any questions, please feel free to let us know during the rebuttal window. Thank you very much! We appreciate your suggestions and comments!

---

> > ### Comment · Reviewer_jcqb · 2021-08-17
> > **Hardware limitations & on-chip training**
> >
> > Thanks to the authors for providing an in-depth response. My points Q.1, Q.2, C.1, C.3, and C.4 have been addressed by the authors. I trust that the authors will implement the detailed comments they have made here into the paper, as they are quite helpful in understanding their contribution.
> >
> > **C.2 Training from scratch**
> >
> > It is still not clear to me if any of the experiments performed were actually done from scratch. The authors mention the `L2ight-SL` series in Table 2 on VGG-8/ResNet-18. To clarify, **was this done with VGG-8/ResNet-18 initialised from zero/random weights on the simulated hardware**? If so, this should be clearly outlined in the paper, as it is very interesting. If not, there is a significant misunderstanding between what the authors call "from scratch" and what the NeurIPS community expects.

---

> > > ### Author Response · Authors · 2021-08-17
> > > **Thank you very much for your comments. Training from scratch in our paper means all weights are zero/random initialized**
> > >
> > > Thank you very much for your comments. This _training from scratch capability_ is the most important contribution of our work. We would like to do the best clarification we can to avoid any misunderstanding.
> > > * The authors would like to clarify that _training from scratch_ mentioned in the paper is without using any pre-trained weights, i.e., all weights are zero/random initialized on the simulated hardware and trained by our proposed subspace learning algorithm. This concept of _from scratch_ is exactly what the machine learning community expects.
> > > * In the paper, the authors highlighted the capability of training from scratch in multiple places:
> > >   * Our Figure 2 shows that our subspace learning stage can be directly used to train ONNs on chip without pre-trained weights.
> > >   * Figure 12. Considering the scale of the plots, we cut all curves in order to highlight the difference when the accuracy is around 90%. But all accuracy curves are actually starting from 10%.
> > >   * In Figure 13, The left-most point represents _no mapping from pre-trained weights_, i.e., training from scratch.
> > > * Our capability to use first-order methods to train ONNs _on hardware from scratch_ is the ultimate source of our claimed 1000x scalability improvement. This scalability does not depend on any off-chip pre-training. Our claims are experimentally supported by all data related to the L2ight-SL series in Table 2.
> > > * The author will give clear explanations to clarify what _training from scratch_ means in the revision so that it will not cause any misunderstanding of our most important contribution. Thank you very much for your great suggestions. It really helps the NeurIPS community better understand the contribution of our paper.
> > >
> > > We wish that our response has addressed your concerns, and turns your assessment to the positive side. If you have any further questions, please feel free to let us know during the rebuttal window. Thank you very much! We appreciate your suggestions and comments!

---

> > > > ### Comment · Reviewer_jcqb · 2021-08-24
> > > > **Rating increased to an accept (7)**
> > > >
> > > > Thank you very much for your detailed answer and your patience. Accordingly, **I have increased my rating to a 7 (accept)**.
> > > >
> > > > This addresses all of my points. Just a note: the *L2ight-SL* name in Table 2 should really be clarified to make it clear this refers to something trained from scratch. It will help readers who want to get a quick overview of the paper just from titles, tables, and figures.

---

> > > ### Author Response · Authors · 2021-08-22
> > > **A Kind Reminder of Our Clarification to your Concerns Regarding 'Training from Scratch'.**
> > >
> > > Thanks a lot for your careful reading of our response and those valuable follow-up questions. This is a kind reminder that **we have a clarification attached** under this thread. Hope we have made it clear that **we do have shown experiments on VGG-8/ResNet-18 with random initialization on the simulated hardware** in Table 2.
> > >
> > > We will clearly outline and highlight this point in the revision. We wish that our response has addressed your concerns, and turns your assessment to the positive side. If you have any questions, please feel free to let us know during the rebuttal window. Thank you very much!

---

### Decision · Program_Chairs · 2021-09-27

**Decision:**

Accept (Poster)

**Comment:**

The paper proposes a calibration and training method for optical neural networks, a possible alternative to the curent implementation in silico. The main topic of the paper is therefore on hardware, and optical implementation of machine learning.  The presented approach here is suited for Mach Zhender interferometers arrays. The framework supports on-chip learning by fine-tuning the implemented weights to compensate for manufacturing imperfections in the chip.

The consensus among reviewer is that this was a very original paper, discussing new direction on the hardware implementation of machine learning algorithm that could be important in the future. The paper was found to be well organised,  and discuss well the state of the art. The new proposed method was found to be clearly described, and the performances to be well backed through simulations. Expert reviewers in both optics and ML assed that the work was interesting and promising.

In the initial round of reviewal: the lack of discussion of the limitations of the framework proposed was pointed out. After rebuttal and discussion between the authors and the reviewers, this was however addressed to the satisfaction of the reviewer.

In conclusion, the proposed approach, L2ight, was thus found to be a significant improvement over the state-of-the-art, building upon and improving previous works clearly. Given the importance of the topic of hardware implementation of machine learning, the area chair thus recommend acceptance to Neurips.